# Climate or land cover variations: what is driving observed changes in river peak flows? A data-based attribution study

Jan De Niel[1], Patrick Willems[1]

[1]KU Leuven, Hydraulics Section, Department of Civil Engineering, Kasteelpark Arenberg 40, 3001 Leuven, Belgium

5 *Correspondence to*: Jan De Niel (jan.deniel@kuleuven.be)

**Abstract.** Climate change and land cover changes are influencing the hydrological regime of our rivers. The intensification of the hydrological cycle caused by climate change is projected to cause more flooding in winters and land use/land cover changes could amplify these effects by e.g. a quicker runoff on paved surfaces. The relative importance of both drivers, however, is still uncertain and interaction effects between both drivers are not yet well understood.

10 In order to better understand the hydrological impact of climate variability and land cover changes, including their interaction effects, we fitted a statistical model to historical data over three decades for 29 catchments in Flanders, covering various catchment characteristics. It was found that the catchment characteristics explain up to 18% of changes in river peak flows, climate variability 6% and land cover changes 8%. Steep catchments and catchments with a high proportion of loamic soils are subject to higher peak flows and an increase in urban area of +1% might cause increases in river peak flows up to +5%. 15 Interaction terms explain up to 32% of the peak flow changes, where flat catchments with a low loamic soil content are more sensitive to land cover changes with respect to peak flow anomalies.

## 1 Introduction

Our environment has undergone unprecedented changes over the past decades, and it is very likely that further changes will take place in the coming decades. With respect to the climate system, increases in frequency, intensity and/or amount of heavy 20 precipitation are globally reported for the majority of the land areas (IPCC, 2014); for Flanders (Belgium) in particular, extreme precipitation might increase with +50% in winter and +100% in summer by the late 21st century (Tabari et al., 2015). With respect to the built environment, the world continues to urbanize, with nowadays 55% of the world's population living in urban areas. This is in shear contrast with 1950, where only 30% of the world's population was urban (United Nations, 2018). For Flanders, this is translated into a 300% increase in built-up area over the past four decades (Poelmans, 2010; Ruimte 25 Vlaanderen, 2017).

Changes in climate and urbanization both cause changes in the hydrological regime of catchments in general and changes in flood frequencies in particular. Here, we aim to attribute observed changes in river peak flows to drivers related to the climate and to a changed land use/land cover. Previous attribution studies related to trends in flood hazards faced several challenges. These were, among others, summarized by Merz et al. (2012). The attribution process typically involves two steps: detection 30 of change and attribution of that change to its various drivers. In the first step, the detection of change is often challenging: the signal of flood time series (or river peak flows in general) typically shows a high natural variability, with a low signal-to-noise ratio. Moreover, floods form part the larger hydrological system and, as such, show a quite complex behavior. With respect to the attribution issue in the second step: in a complex hydrological system, different drivers act in parallel, with interactions between them. The integral response of the system to all these drivers and interactions governs the changed hydrological 35 behavior. And, finally, the power of attribution studies often lies in a deep process knowledge related to the proposed driver-effect mechanisms (Hegerl et al., 2010); unfortunately, knowledge on some driver-effect mechanisms is still limited (Blöschl et al., 2007; Dey and Mishra, 2017; Van Loon et al., 2016; Merz et al., 2012).

On the driver-effect mechanism between climate variability and river peak flows, many studies have shown there is a link between weather types and flooding, sometimes through the intermediate variable of precipitation (Brisson et al., 2011; Hirschboek, 1991; Mediero et al., 2015; De Niel et al., 2017; Pattison and Lane, 2012; Pfister et al., 2004; Prudhomme and Genevier, 2011; Santos et al., 2015; Smith et al., 2011; Wilby and Quinn, 2013). For the area of Flanders, westerly atmospheric fluxes would, in general, cause an increased winter precipitation amount and intensity, leading to increased river peak flows (Brisson et al., 2011; De Niel et al., 2017; Willems, 2013).

On the driver-effect mechanism between land use/land cover and river peak flows, most studies hypothesize that deforestation and increased urbanization cause increased surface runoff. (Bronstert et al., 2002; Cheng and Wang, 2002; Cuo et al., 2009; Galster et al., 2006; Hamdi et al., 2011; Hundecha and Bárdossy, 2004; Miller et al., 2014; Misra, 2011; O'Driscoll et al., 2010; Pfister et al., 2004; Poelmans et al., 2011; Reynard et al., 2001; Siriwardena et al., 2006; Trudeau and Richardson, 2016; Zope et al., 2016). Most of these studies look at the integral response of the catchment due a changed land use/land cover, and do not aim to attribute the changes to the specific type of changes that occur: e.g. an increase in settlement at the expense of agricultural land. Also, a lot of uncertainty remains, mainly because of the heterogeneity of hydrological responses and the scale of the river basin/catchment considered (see e.g. Zhang et al. (2017).

Next to the independent driver-effect mechanisms of climate variability on river peak flows, and land use changes on river peak flows, both drivers should be analyzed jointly in a multiple-driver attribution study (e.g. Hall et al., 2014; Merz et al., 2012). As an example, for the Meuse river, it was concluded that changes in flood frequency and magnitude over the past century could mainly be attributed to climate variability rather than to deforestation and urbanization (Tu et al., 2005). Similarly, for the Rhine and Meuse basins, increased flooding probability was found to be correlated to an observed increase in westerly atmospheric fluxes (causing an increase in winter precipitation amount and intensity) and not to observed land use changes (Pfister et al., 2004). For a smaller catchment such as the Grote Nete (385 km$^2$, located in the North-East of Flanders), and for the future conditions, both climate change and urban growth are projected to have a considerable impact on river peak flows (Tavakoli et al., 2014; Vansteenkiste et al., 2014).

With this paper, we investigate the (relative) importance of climate variability and land cover changes related to changes in river peak flows, based on 29 catchments throughout Flanders. For the historical dataset covering the past three decades (Section 2), a data-based approach is followed where peak flow anomalies are explained based on a set of maximum 24 drivers. These drivers are grouped into three categories: catchments specific drivers, climate variability and land use/land cover changes. A model is built based on panel data regression, with a top-down approach (Section 3). Results are presented in Section 4 and overall conclusions are given in Section 5.

## 2    Study area and data

For this case study, 29 catchments are selected, evenly spread across Flanders, the Northern part of Belgium (Figure 1).

Flanders, with 6.4 million inhabitants, covers around 13,500 km$^2$. The coastal area in the North-West of the region mainly consists of sand dunes and clayey alluvial soils in the polders. The central area mainly consists of loamic soils and ranges between 0 and 10 mTAW, with mTAW the height, in meters, above the local mean sea level. The North-Eastern part, known as the Campine region, has sandy soils at altitudes around 30 mTAW. The Southern part with silty soils has low hills op to 150 mTAW. The maximum height is 288 mTAW in the South East. The DTM in Figure 1 was taken from the Digital Elevation Model Flanders (https://overheid.vlaanderen.be/producten-diensten/digitaal-hoogtemodel-dhmv). Soil texture data were obtained from the Flanders underground database (www.dov.vlaanderen.be).

Flanders has a maritime climate (*Cfb*, according to the Koppen climate classification), with average temperatures of 3 °C and 18 °C in January and July, respectively. There is a small gradient present with lower temperatures in the South-East (annual average of 10 °C) towards higher temperatures in the North-West (annual average of 11 °C) (based on the period 1981-2010); the average temperatures in Flanders, further, has been rising over the past 30 years with 1 – 1.5 °C. Average evapotranspiration was 540 mm/year in 1980 and rose to 625 mm/year in 2010. Yearly precipitation varies between 600 mm/year to 1000 mm/year, with little variation throughout the year, and little spatial differences (Brouwers et al., 2015).

Twenty-nine catchments were selected based on a minimum of 20 years of available discharge data (www.waterinfo.be). Some of the main characteristics of these catchments are listed in Table 1. Further, Figure 2 and Figure 3 show details on land cover and soil texture of these catchments, respectively. For land cover, the 30 classes from the ESA CCI Land Cover project (www.esa-landcover-cci.org) were regrouped into the 6 IPCC land categories, i.e. cropland, forest, grassland, wetland, settlement and other land. This was done in order to reduce the total degrees of freedom for this study. Soil texture is obtained from www.dov.vlaanderen.be; 3 dominant soil textures (Arenic, Loamic and Siltic) cover 99.3% of the total area of the selected catchments. Therefore, further in this study, only these 3 dominant soil textures were taken into account.

Climatic conditions in the past are based on the NCEP/NCAR reanalysis data, available online through https://www.esrl.noaa.gov/psd/ (Kalnay et al., 1996).

## 3 Methods

### 3.1 General

The aim of the study is to find the main drivers behind changes in river peak flows. Therefore, the hourly discharge series of each catchment is first transformed to peak flow anomalies (Sect. 3.2). Then, possible drivers are derived from the data introduced in Sect. 2 and further split into separate categories, see Sect. 3.3. Finally, a regression model is fitted to the data (Sect. 3.4).

### 3.2 Peak flow anomalies

The methodology to estimate peak flow anomalies is schematized in Figure 4. The hourly discharge data (Figure 4a) is first split into independent events and extremes are extracted (see Figure 4b), based on the method proposed by (Willems, 2009). Empirical probabilities (or equivalent return periods) are assigned to these extremes, based on the full time series (reference period) on the one hand, and based on subsets of extremes in subperiods/blocks of 10 years length on the other hand (Figure 4c). The quantiles in a particular subperiod/block are then compared with the corresponding quantiles based on the reference period and the ratio of these two empirical quantiles defines an anomaly factor (Figure 4d). Finally, per subperiod/block of 10 years, all anomaly factors corresponding to a return period larger than one year are averaged in order to get one value per subperiod of 10 years (Figure 4d). As such, one can plot and/or investigate peak flow anomalies for a given catchment over time (Figure 4e). Note that, when investigating these anomalies over time, a detected signal is only considered robust if it persists for a period longer than the selected block period (here: 10 years). If, e.g. an increased anomaly is found for 4 consecutive years and afterwards falls back to the values prior to this increase, this increase is only an artefact of the anomaly method.

### 3.3 Possible drivers

The data introduced in Sect. 2 generally relate to one of the following three categories: catchment characteristics (*CAT*), climate variability (*CLIM*) and land cover changes (*LULC*).

Catchment characteristics are considered time invariant in this study and are derived from following sources: digital terrain model (DTM) with a spatial resolution of 100m x 100m, river map and soil texture. From the DTM, the river map and locations of the outlet stations, catchment delineations are defined. Further, based on the DTM, the slope in the catchment is calculated, as well as the average slope over the whole catchment. A river density is defined as the ratio of total river length in the catchment over the total area of the catchment. Finally, the relative area of the soil textures are being used in the further analysis. For these soil textures, Arenic, Loamic and Siltic were found to cover 99.3% of the area of Flanders; and when Arenic is seen as the complement of (Loamic + Siltic), only two variables remain to describe soil textures. The absence of an explicit variable Arenic is compensated through the constant $\alpha$ in the model (see Section 3.4.1).

Climate variability is derived from the NCEP/NCAR reanalysis data (Kalnay et al., 1996). Here, weather types are derived based on the daily mean sea level pressure from this reanalysis dataset. Different classification methods exist (Philipp et al., 2010); here, the Jenkinson Collison system (Jenkinson and Collison, 1977), a modified version of the Lamb-weather type classification method (Lamb, 1972) is used to convert sea level pressure into one of 28 weather types. These 28 weather types are reduced to 11 by combining all types with the same directional component (see also e.g. (Demuzere et al., 2009)) and further reduced based on the link between river peak flows and weather types (De Niel et al., 2017). The remaining groups of weather types are: W; (NW, N), (NE; E; SE), (S; SW); U; C; A, with N, E, S and W referring to wind directions, C and A to cyclonic and anticyclonic atmospheric patterns, respectively, and U to an unclassified weather type. This reduction aims to limit the degree of freedom in the final model. In the further analysis, relative frequencies of these daily weather types are considered, based on a rolling window of 5 years (Figure 5), and U is considered as the complement of the other groups of weather types.

Six IPCC land categories (settlement, agriculture, grassland, forest, wetland and other area) are taken into consideration as possible drivers for this study. It is seen that the maximum proportion of Wetland and Other area in the considered catchments is equal to only 0.2% and 1.5% respectively. Therefore, these *LULC*-classes will further not be taken into account. In addition, the *LULC* class Grassland is considered as the complement of (Forest + Agriculture + Settlement). Because the *LULC* database does not show any significant changes after 2005 (Figure 2), the analysis is limited to 1992-2005.

Table 2 summarizes the possible drivers considered in this attribution study.

### 3.4    Regression model

#### 3.4.1.    Panel data analysis

A model is built with the techniques and ideas of panel analysis, which is widely used in social sciences, epidemiology, and econometrics where two dimensional data is analysed. Typically, in those sectors data is collected over time and over the same individuals. Here, the two dimensions are space and time – input data can show only a temporal variation (e.g. climate data), only a spatial variation (e.g. soil texture), or a combination of both (e.g. LULC). Note that, typically, climate data does show a spatial variation as well. However, we assume the area of Flanders to be homogeneous with respect to the considered climate data.

The typical panel data regression model can be described as follows:

$$y_{it} = \alpha + \boldsymbol{\beta} \boldsymbol{X}_{it} + \epsilon_{it}, \tag{1}$$

with $y$ the output of interest, $i$ the individual (or catchment), and $t$ the time; $\alpha$ and $\boldsymbol{\beta}$ are constants, of dimension (1 x 1), and (1 x n) respectively, with $n$ being the number of inputs/observations considered. Note that both $\alpha$ and $\boldsymbol{\beta}$ are catchment independent, as no index $i$ appears here. $\boldsymbol{X}$ represents the input/observations as explanatory variables, with dimension (n x 1) for each individual (or catchment) at a particular time $t$ and $\epsilon$ is an error term. In this study, the output of interest is peak

flow anomaly, and inputs can be split into three categories: catchment specific characteristics *CAT*, climate variability indicators *CLIM* and land cover *LULC*, as described in Table 2. As such, $\boldsymbol{X}_{it}$ from Eq. (1) becomes:

$$\boldsymbol{X}_{it} = (\boldsymbol{CAT\ CLIM\ LULC})_{it}^T, \tag{2}$$

with superscript T indicating the transpose of a matrix. Next to the linear model (Eq. (1)), combined effects of (changes in) observed variables might also play a role in explaining the changes in the output of interest.. Therefore, an interaction term is added to the model:

$$y_{it} = \alpha + \boldsymbol{\beta X}_{it} + \boldsymbol{\rho\ X}_{it}^T\ \boldsymbol{X}_{it} + \epsilon_{it}. \tag{3}$$

The interaction matrix $\boldsymbol{\rho}$ is of dimension (n x n) and is constant, hence time and catchment independent. This matrix is a strictly upper triangular matrix, meaning all entries on and below the main diagonal are all equal to 0. Furthermore, for our study, we added the restriction that there cannot be any interaction between explanatory variables from within the same category: e.g. $\rho_{area,slope} = 0$.

### 3.4.2. Model building

Model building happens based on a top down approach. Starting from a simple constant model, with $\boldsymbol{\beta = 0}$ and $\boldsymbol{\rho = 0}$, explanatory variables are added to the model based on changes in the value of the Bayesian information criterion *BIC* (Kass and Raftery, 1995). BIC is a general criterion for model selection, where models with the lowest BIC are preferred. It takes into account the likelihood of a model, the sample size and the number of parameters estimated by the model. In a first step, only the linear model (Eq. (1)) is considered. Once the linear model is fixed, interaction terms are added in a similar way. Note that we only consider interactions between variables present in the linear model. E.g. if $\beta_{arenic}$ would be equal to 0 in the linear model, then all $\rho_{arenic,X}$ in the model including interaction terms are, a priori, set equal to 0.

In order to build a robust model, 100 linear models are tested based on (100 times) 20 random calibration catchments. Based on this set of 100 models, significant variables are selected, i.e. variables which appear in the majority of the models.

## 4 Results and discussion

### 4.1 Final model

The final model has 26 terms in 9 predictors (see Table 3). During model building, it was decided to not further consider following variables (Figure 6):

- Catchment characteristics: Area;
- Climate variability: W; (NW, N); (NE,E,SE); A and U.

The catchment area does not have a significant contribution in explaining observed peak flow changes. Furthermore, when including interaction factors between catchment area and the other variables, model performance did not improve (not shown). This might seem surprising at first, since Bloschl et al. (2007), among others, hypothesize that land use impact on hydrological response is depending on the catchment scale. However, all selected case studies are considered to be of the same scale, despite the differences in catchment area and thus, the hypothesized effect of catchment scale on land use impacts is not applicable here.

One should be careful when interpreting the coefficients from the final model in Table 3. E.g. the coefficient of Settlement in the final model is equal to -3.04. At first sight, an increase of settlement would thus correspond with a decrease of peak flow anomaly. However, the interpretation of the coefficients is more complex:

- An increased Settlement also impacts the interaction effects, and the coefficient becomes: (-3.04 – 0.85*Slope + 6.47*Loam + 17.85*Settlement);
- An increased Settlement means that Agriculture (13.08) and/or Forest (3.71) might decrease – and there again, the interaction effects of Agriculture and Forest come into play.

The model is able to explain 60% of the changes in river peak flows over time (Figure 7). This performance is further broken down into linear effects of the three separate groups and their interactions (Figure 8). Linear effects (28%) are found to be of equal importance as interaction effects (32%). Within the linear effects, catchment characteristics are most important as they explain the highest portion (18%) of the river peak flow changes, followed by land use/land cover (8%) and climate variability (6%). These percentages were obtained by only considering the models that include the variable considered. Note that 18% +

8% + 6% is only slightly larger than 28%, which is due to a small interdependency between land use/land cover, soil texture and catchment slope.

Observed peak flow anomalies in catchments L07_289 (Mark at Viaene) and L08_233 (Zuunbeek at Sint-Pieters-Leeuw) have a bad correspondence with their modelled results (Figure 7). The Mark catchment has a long history of flooding – as from the 2000s, the local authorities have installed several mitigation measures (hydraulic structures, retention basins etc.), effectively

decreasing the flood risk. This is also visible in the observed peak flow anomaly. However, the regression model used in this study cannot capture such management changes. Further, for the Zuunbeek catchment at Sint-Pieters-Leeuw, increased peak flow anomalies are observed as from the middle of the period. This is due to the extreme flood season in the winter of 2001-2002 where 7 events were observed with peak discharges exceeding 6 $m^3$/s, corresponding to an empirical return period larger than 1 year, based on data between 1978 and 2016.

**4.2     Effect of single drivers**

Firstly, the dependency of peak flow anomalies to catchment characteristics is investigated. This is done by only considering those factors of the model, solely consisting of catchment characteristics. It is seen, from Figure 9, that peak flow anomalies go up with an increased slope, lower proportion of loamic soil textures and higher proportions of siltic soil textures in the catchments. With respect to density, the results show less clarity.

These findings correspond to an analysis done on the potential runoff coefficient as used in the hydrological model Wetspa (Liu and De Smedt, 2004). The potential runoff coefficient of a catchment is defined as the ratio of runoff volume to rainfall volume. A simple and practical technique was developed in Wetspa to estimate this runoff coefficient as a function of land use, soil texture and slope, based on reference values from literature (Browne, 1990; Chow et al., 1988; Fetter, 1980). See e.g. Figure 10 for potential runoff coefficients in Wetspa for different combinations of LULC, slope and soil texture. Note that they

use slightly different LULC classes, but these differences are insignificant for the purpose of this discussion. From a hydrological point of view, relative changes in potential runoff coefficient can serve as a proxy for peak flow anomalies. As such, findings with respect to the potential runoff coefficient from Wetspa can be related with the conclusions based on Figure 9:

- Figure 10a shows that potential runoff coefficients increase, with increasing slope. Moreover, the rate of this increase

is lower for higher slopes. This corresponds with our findings on catchment slope.
- Figure 10b and c show that potential runoff coefficients are generally lower for a loamic soil texture compared with a siltic soil texture. This corresponds with our findings on the impact of soil texture classes.

Secondly, with respect to the climate system, it was seen that the relative frequencies of S+SW, combined with the relative frequencies of A give the most information to the model explaining peak flow anomalies (Table 3 and Figure 6). This, however,

does not mean that the hydrological cycle is mostly/only depending on these weather types. Correlations exist between the

various weather types; for example frequencies of anticyclonic and cyclonic weather show a negative correlation of -0.79. Because of these correlations, we do make any statements on the effect of increasing/decreasing frequencies of S+SW or A on peak flow anomalies.

Finally, based on the model, the overall impact of increased urbanization can be investigated. This is done by changing, for each catchment, 1% of the total area from settlement to forest, grassland and agriculture, respectively. This results for most catchments in increased peak flows (Figure 11), with disappearing grassland in favour of settlement area causing the biggest changes. These results are in line with (Hundecha and Bárdossy, 2004), who found an increase of 7 – 10% in river peak flows for a 15% increase in urban area at the expense of agricultural land. The strongest changes were found for catchments L01_491, L01_492, L01_496 and L05_404. These catchments are all quite flat and have a high proportion of loamic soil texture. This finding will further be discussed by investigating interaction effects below.

### 4.3    Interaction effects

The total amount of interaction effects (32%) is largely carried by three terms only: interaction between LULC and soil texture classes (% loamic) (10%), between LULC and slope (6%) and between soil texture classes (% loamic) and slope (6%) (see Figure 12):

- Figure 12a shows effects of LULC changes on peak flow anomalies as a function of the slope (three particular slopes are shown: flat (0.40), medium (2.83) and steep (5.26)). Note that this graph was obtained by averaging out effects of other predictors and, as such, the absolutes values of the effects should be interpreted carefully. For the purpose of interaction effects, results of Figure 12 should be interpreted in a relative way. It is seen that, with increasing slope, the effect of LULC changes on peak flow anomaly goes down. A steeper slope typically results in increased peak flows but the LULC changes influence these peak flow anomalies in a lesser degree, compared with more flat catchments. Note that, although different in magnitude, these trends are consistent for each LULC class.
- Similar to this interaction between slope and LULC, catchments with a low proportion of loamic soil textures are less influenced by LULC changes with respect to peak flow anomalies, compared to catchments with a high proportion of loamic soils (Figure 12b). Again, trends are consistent for each LULC class.
- And finally, the catchment slope has a larger effect with respect to peak flow anomalies in catchments with a high proportion on loamic soil textures, compared to catchment with a lower proportion on loamic soil textures (Figure 12c).

Comparison with the analysis on the potential runoff coefficient from Wetspa (Figure 10) learns the following on the three main interaction effects:

- Slope and LULC: One can see in Figure 10a that the range of potential runoff coefficients between the four LULC classes is significantly larger at a near-zero slope, compared with a slope of 100%. In other words, relative changes in the potential runoff coefficient with changing LULC are smaller for catchments with a steeper slope.
- Soil texture and LULC: For catchments with a pure loamic soil texture, the potential runoff coefficient at a near-zero slope increases with a factor 4.4 from a forested area (0.14) to mixed urban (0.62). For catchments with a pure siltic soil texture (thus, with a very low contribution of loam), this is only a factor 3.1 (0.21 vs. 0.66) (Figure 10b). In other words, loamic catchments are more sensitive to LULC changes with respect to potential runoff coefficients.
- Soil texture and slope: For catchments with a pure loamic soil texture, the potential runoff coefficient in forested area increases with 42% between a slope of 1% (0.14) and 5% (0.20). For catchments with a pure siltic soil texture (thus, with a very low contribution of loam), this is only 29% (0.21 vs. 0.27) (Figure 10c). In other words, loamic catchments are more sensitive to the catchment slope with respect to potential runoff coefficients.

Interaction terms between LULC and climatic conditions holds only 2% of explanatory power in the models. Figure 13 shows these minor interactions. Periods in time rich on anticyclonic weather types show a decreased sensitivity on changes in agricultural and forested land, and an increased sensitivity on settlement area. Moreover, a decreased sensitivity to agricultural land is seen for periods rich on S and SW weather types. However, as the confidence intervals for the different climatic conditions overlap in all four cases of Figure 13, these interactions might not be significant.

The remaining interaction terms from Table 3 further explain an additional 8% of the variation in peak flow anomalies. Note that no significant interaction terms were found between catchment characteristics and climate conditions. This would mean that each catchment responds in a similar way to climatic oscillations.

## 5    Conclusion

The regression model is able to explain 60% of the changes in peak flow extremes. For some individual catchments, however, the model is not able to mimic observed step changes, e.g. for catchments L07_289 (Mark at Viaene) and L08_233 (Zuunbeek at Sint-Pieters-Leeuw). For the other 27 considered catchments, the direction and the overall trends simulated by the model are found to be accurate.

With respect to the estimation of the regression model, ideally, one would carry out a split-sample test (in space and in time); however, because of data availability and spatial heterogeneity, this approach would fail in this case. Alternatively, the robustness of the model is tested here by fitting multiple models with different calibration data. It is seen from Figure 7 that this approach results in consistent estimations for the peak flow anomalies – only for catchment L11_518 this consistency was not always found.

It was seen that for these case studies, changes in land cover and climate variability play an equally important role in explaining changes in river peak flows. These effects, however, are of a lower importance than catchment specific factors, such as topography and soil texture: higher peak flow can be expected for catchments with a high average slope, a low proportion of loamic soil texture and high proportion of siltic soil. The high importance of these time-invariant factors (topography and soil texture) indicate that flood response in Flanders is highly catchment specific, and to a lesser degree depending on fluctuations of the climate and land use changes.

Obviously, given the complexity of these environmental systems, the simple linear model will not be able to capture/describe all effects – indeed, it was seen that interaction effects between catchment characteristics, land cover and climate variability are equally important in explaining changes in river peak flows. It was shown that the sensitivity with respect to peak flow changes caused by LULC changes is lower for catchments with a steep slope and a low proportion of loamic soil textures. The model also showed that, for most of the considered case studies, a decrease in forested area to increase settlement area indeed leads to increased peak flows. Moreover, 1% increase in settlement could lead in some cases to a 5% increase in river peak flows. These findings provide important new findings in support of urban planning and flood management. Firstly, the link between slope, soil texture and peak flows can help in developing catchment specific flood management plans. Also, the land use changes should be planned taking catchment characteristics into account since it was shown that land use change impacts on peak flows differ significantly in catchments with different slopes and soil textures.

**Data availability**

All data were obtained via publicly available sources. The DEM was obtained from "Digitaal Hoogtemodel Vlaanderen" (VMM, Watlab and Agiv), available from https://overheid.vlaanderen.be/producten-diensten/digitaal-hoogtemodel-dhmv. The river network and catchment delineation was obtained from "Vlaamse Hydrografische Atlas" (www.geopunt.be). Land cover

was obtained from the ESA CCI Land Cover project (https://www.esa-landcover-cci.org/). Soil texture data is available on www.dov.vlaanderen.be. The NCEP/NCAR Reanalysis data were provided by the NOAA/OAR/ESRL PSD, Boulder Colorado, USA, from their website at https://www.esrl.noaa.gov/psd/. The discharge data were obtained from www.waterinfo.be

## 5 Author contribution

JD and PW worked on the concptualization of the research and jointly developed the methodology. JD curated the data, performed the formal analysis and investigation, under the supervision of PW. JD prepared the visualisation and the initial draft, which was critically reviewed and revised by PW.

### Acknowledgements

The authors would like to thank colleague Els Van Uytven from KU Leuven for deriving the weather types. Also, the organizations VMM, Watlab, Agiv, ESA and NOAA are gratefully acknowledged for making their data publicly available. We would also like to thank the three anonymous reviewers and the editor whose comments and suggestions helped improve and clarify this manuscript.

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

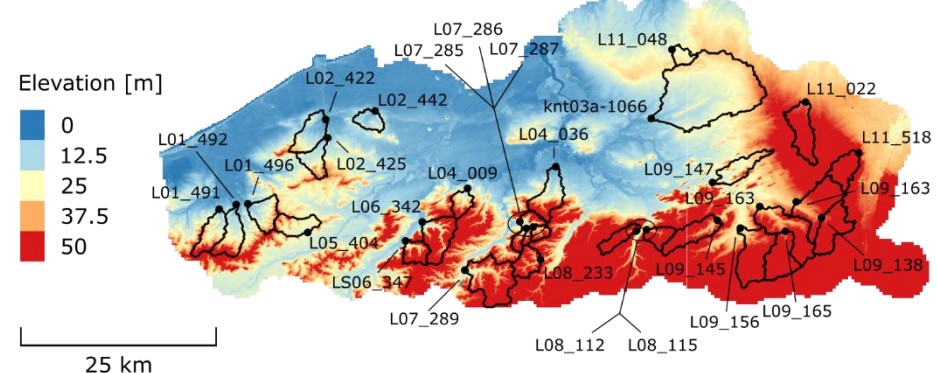

**Figure 1. Selected catchments in the Flanders area of Belgium.**

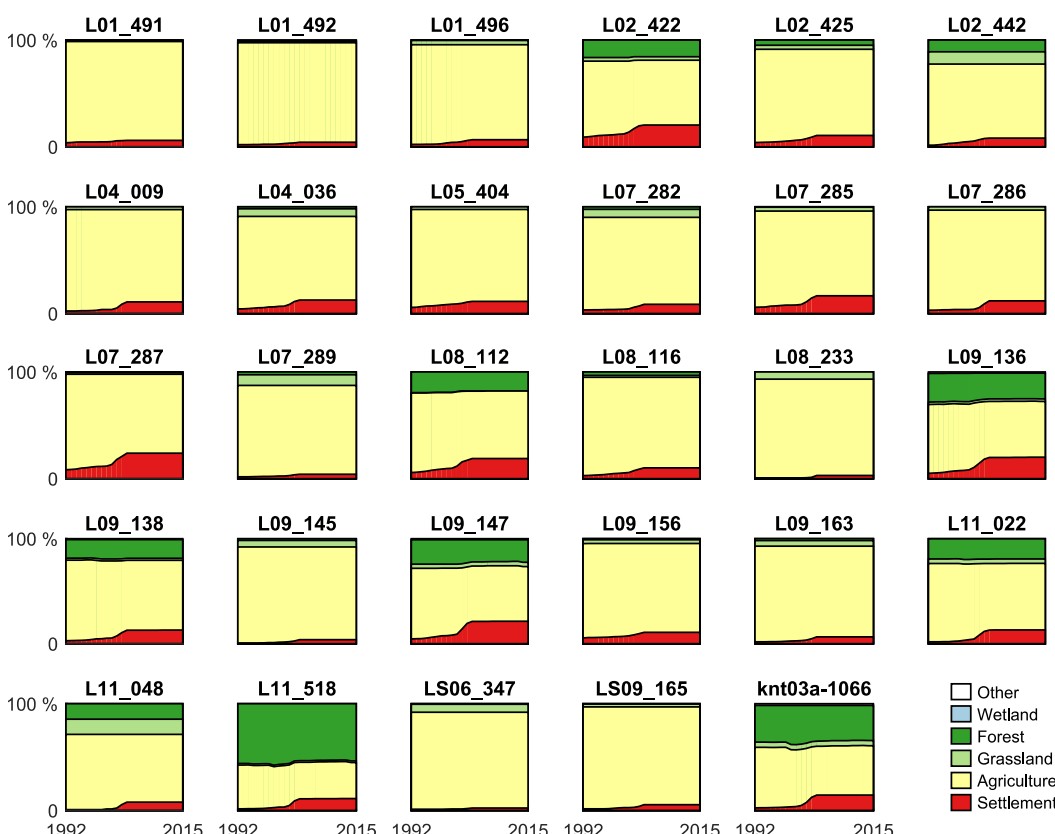

5   **Figure 2. Land cover and land cover changes over time (1992 – 2015) for the selected catchments.**

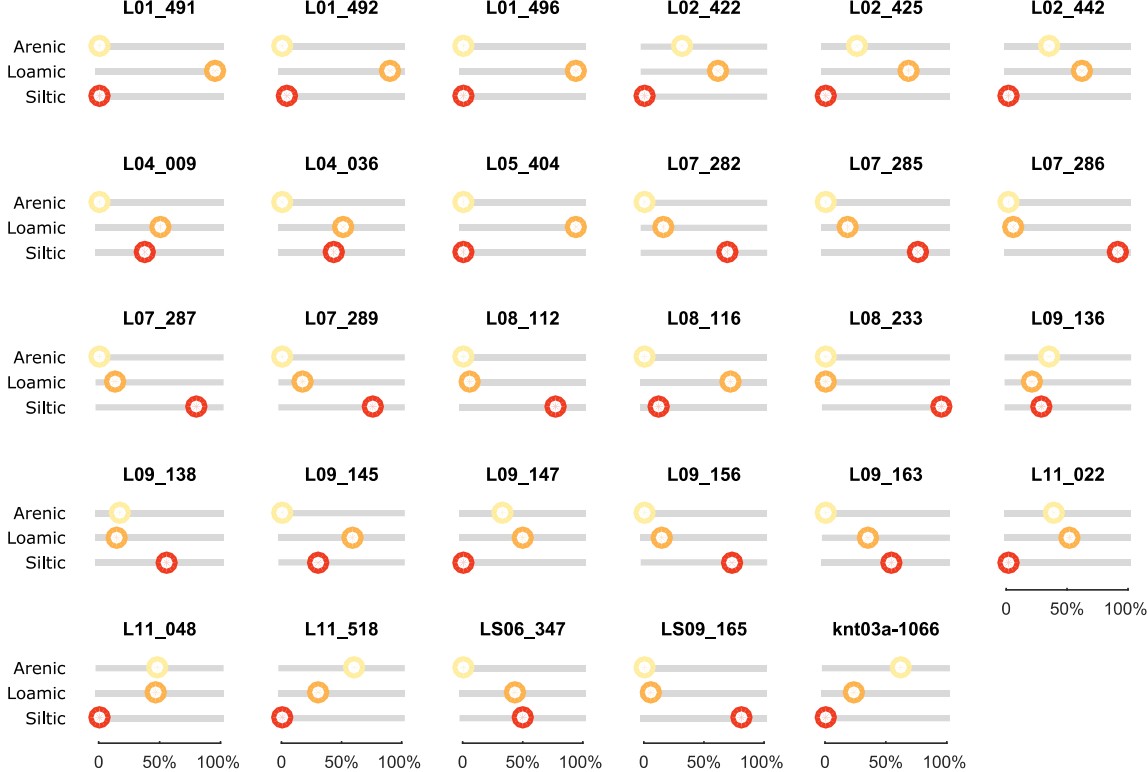

**Figure 3. Relative areas of soil texture (arenic, loamic and siltic) for the selected catchments. Data from: www.dov.vlaanderen.be**

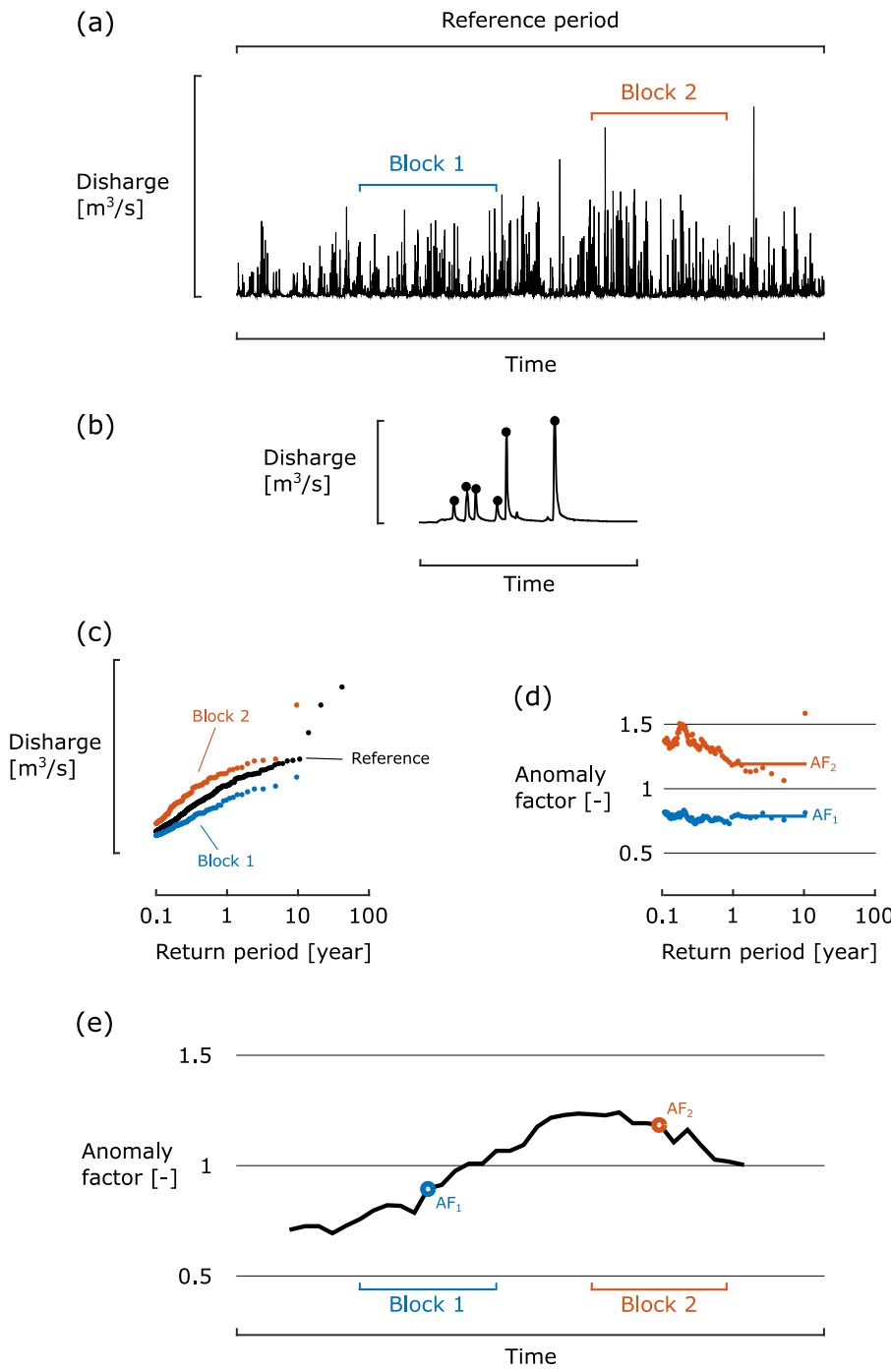

**Figure 4. Schematic overview of methodology to estimate peak flow anomalies for two specific block periods over a reference period.**

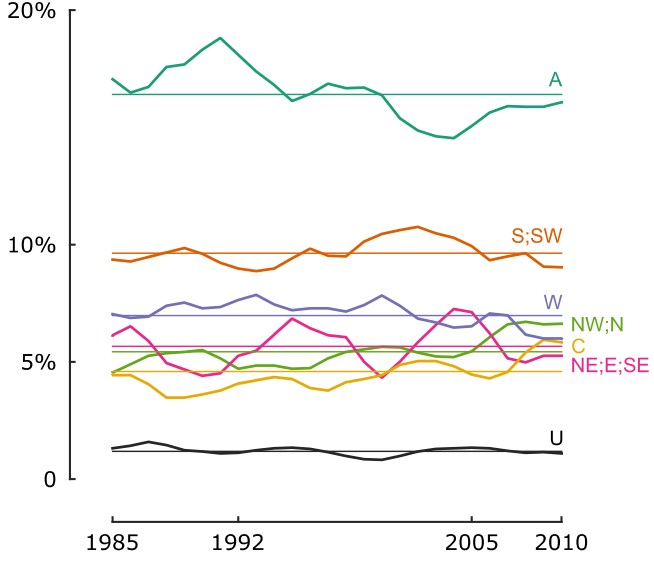

**Figure 5. Relative frequency of Lamb weather types over the years.**

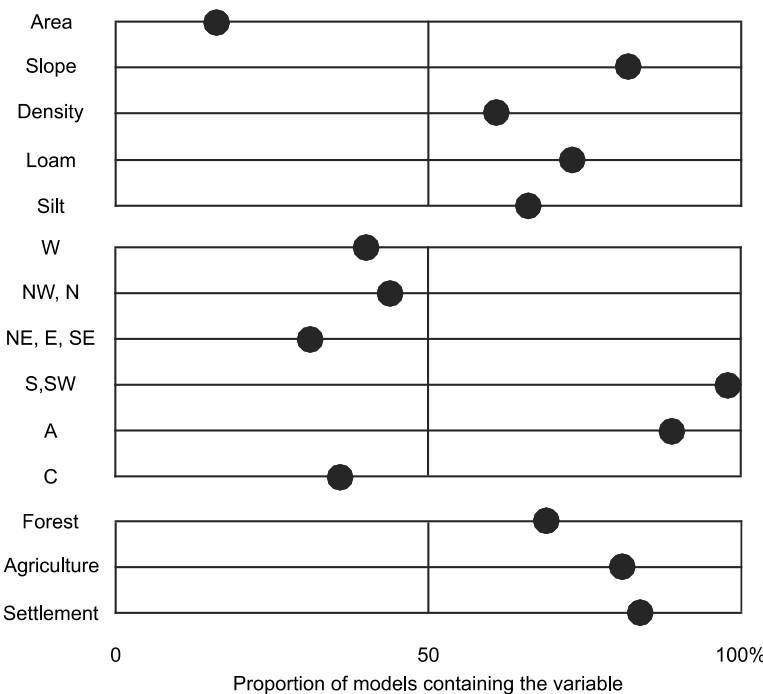

**Figure 6. Variables appearing in >50% of the calibrated models are selected to explain changes in river peak flows.**

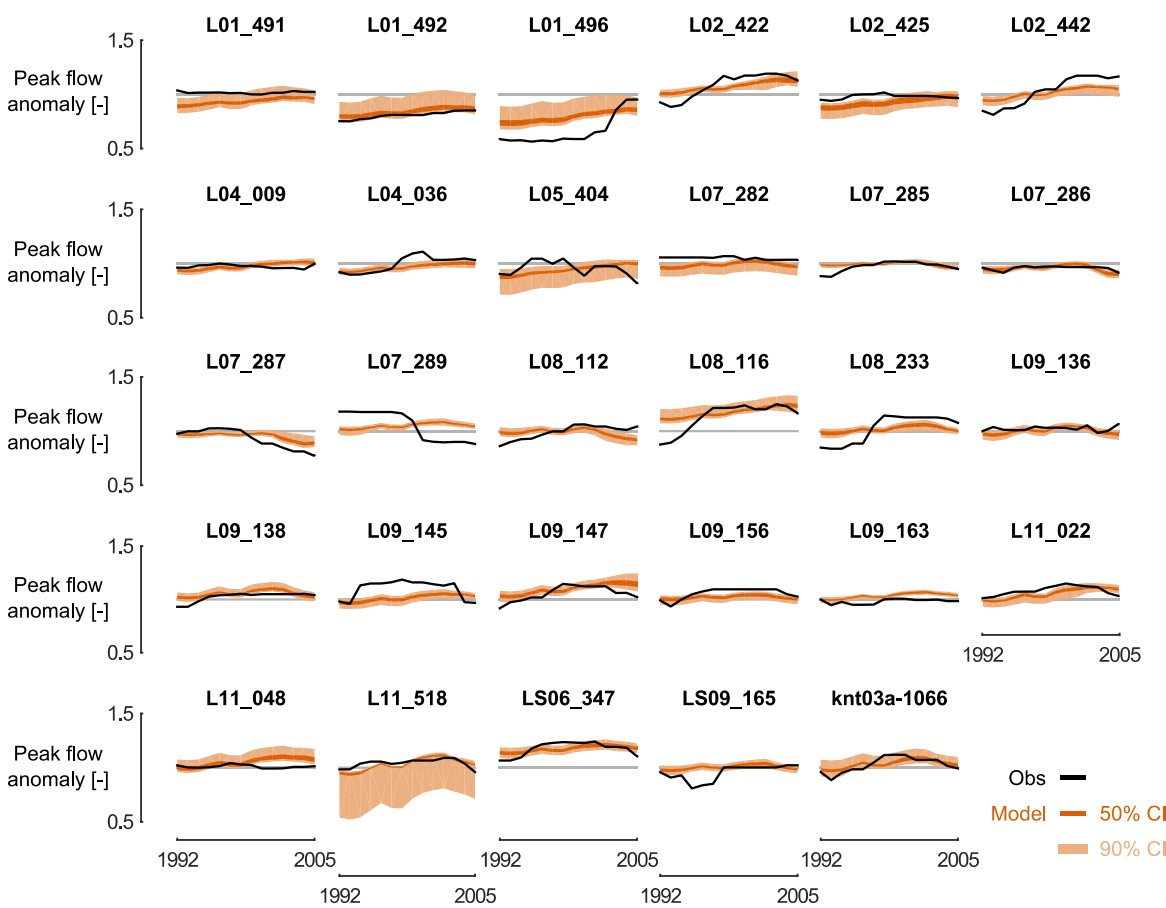

**Figure 7. Regression model combining catchment characteristics, climate variability and land cover changes to explain streamflow variability.**

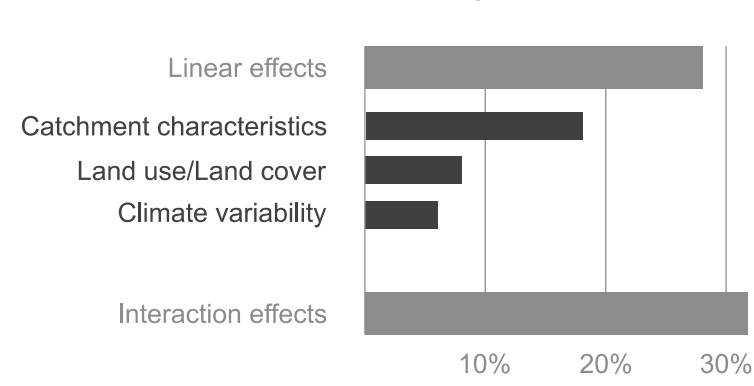

5   **Figure 8. Linear effects and interaction effects between catchment characteristics, climate variability and land cover changes play an equal role in explaining streamflow variability.**

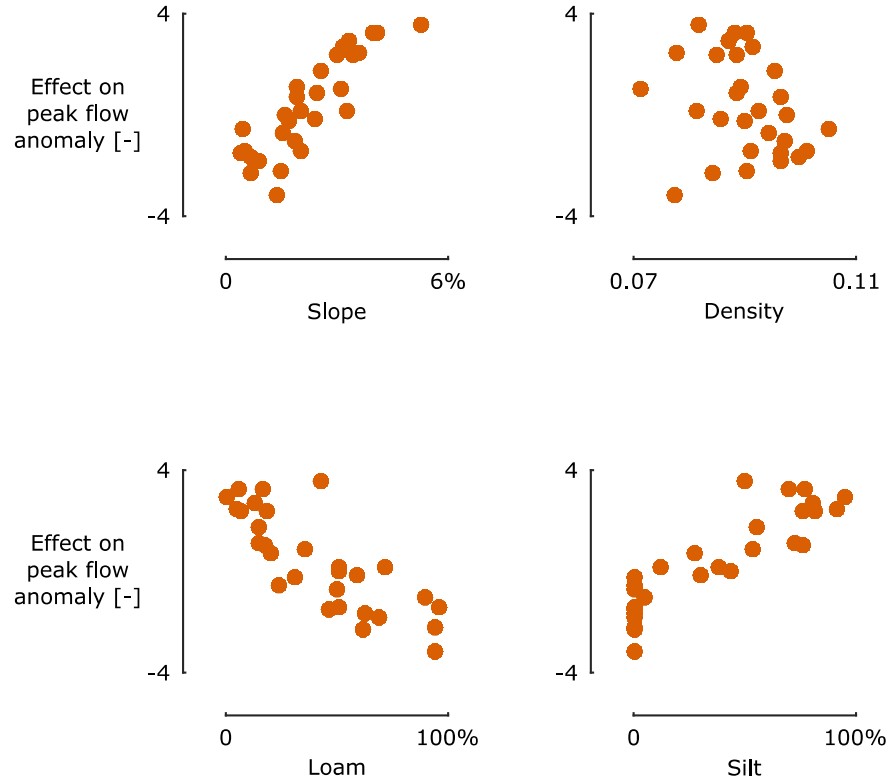

**Figure 9. Effect of catchment characteristics on peak flow anomalies for the 29 selected catchments.**

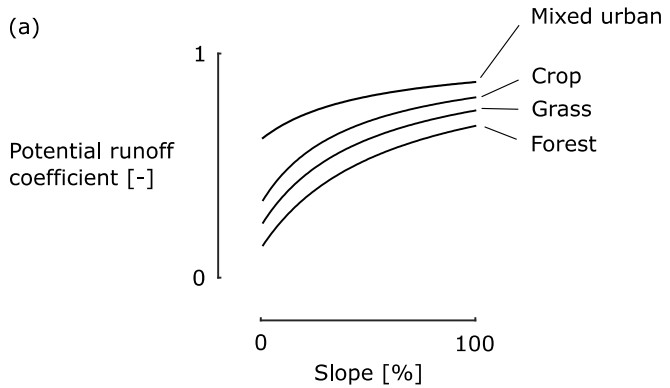

(a)

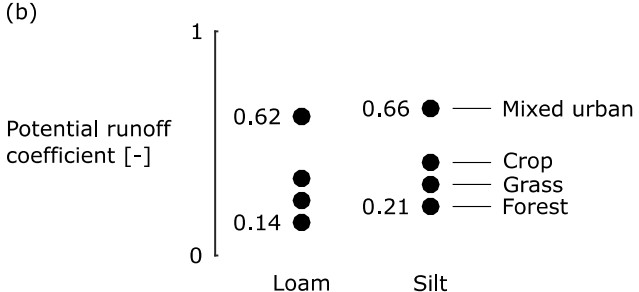

(b)

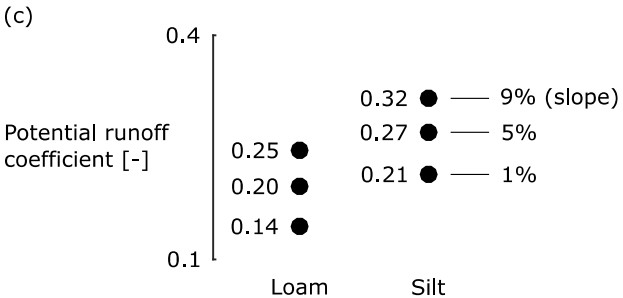

(c)

**Figure 10. Potential runoff coefficient from the Wetspa hydrological model** (Liu and De Smedt, 2004)**, (a) as a function of slope, for different LULC categories (loamic soil texture), (b) as a function of soil texture class for different LULC categories (near-zero slope) and (c) as a function of soil texture class for different slopes (forested area).**

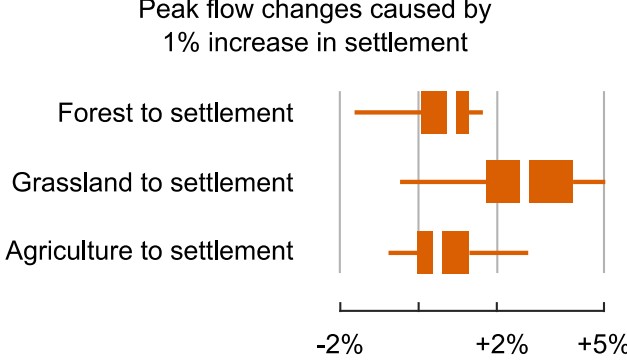

**Figure 11. Peak flow changes by increasing settlement area through decreasing forest, grassland or agriculture.**

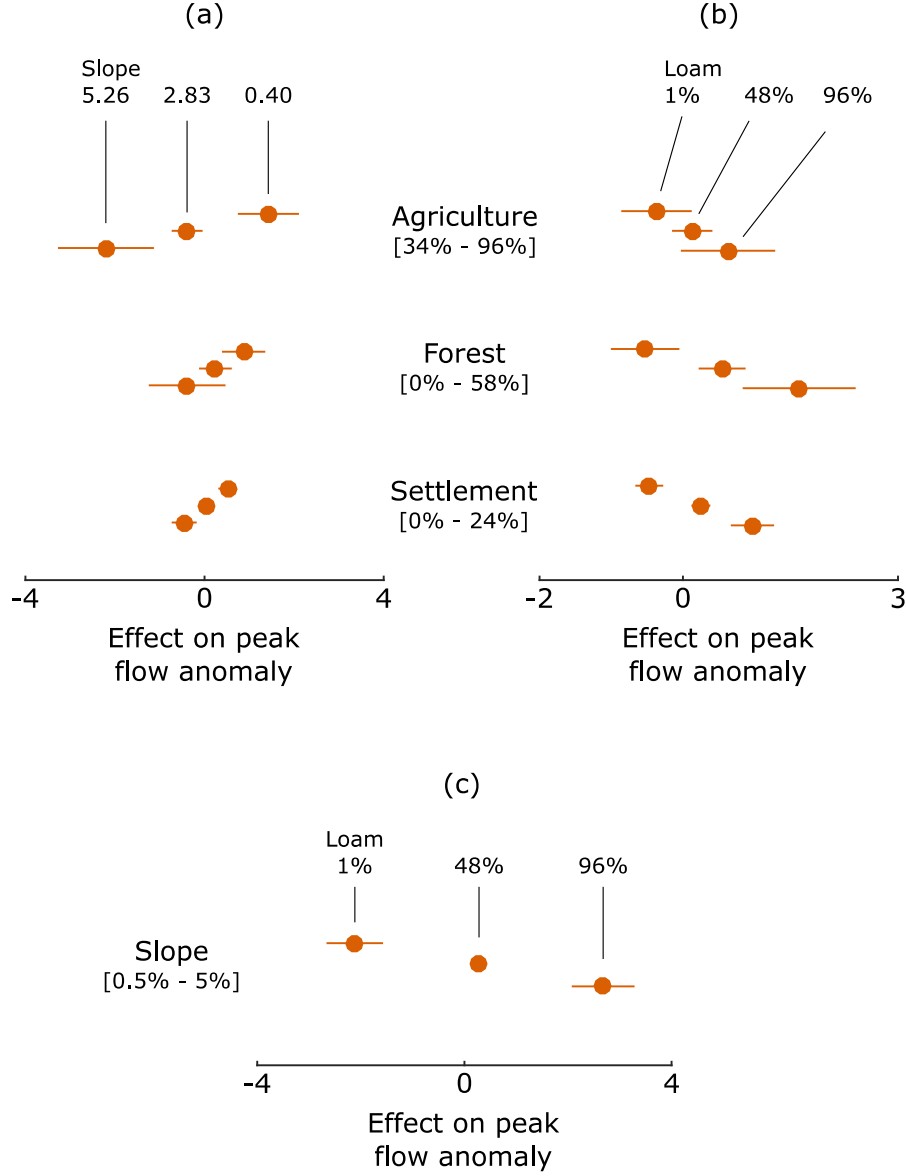

**Figure 12.** Estimated effect on peak flow anomalies from changing (a) slopes and LULC, (b) soil texture (loamic content) and LULC and (c) slopes and soil texture (loamic content), averaging out the effects of the other predictors. Horizontal bars indicate confidence intervals for the estimated effect.

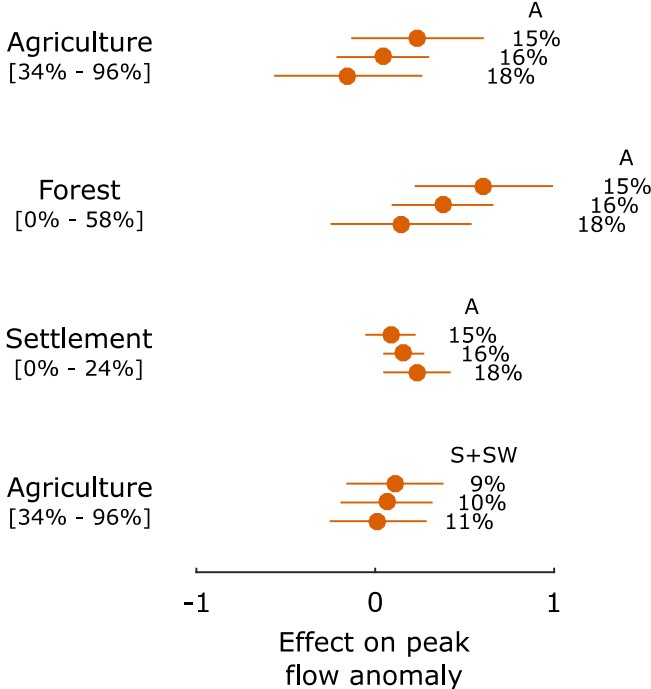

**Figure 13. Estimated effect on peak flow anomalies from changing LULC (Agriculture, Forest and Settlement) and climatic conditions (relative frequencies of weather types S+SW and A), averaging out the effects of the other predictors. Horizontal bars indicate confidence intervals for the estimated effect.**

**Table 1. Main characteristics of the selected catchments.**

| Id. | Outlet station | River | Area [km$^2$] | Period | | # years |
|---|---|---|---|---|---|---|
| knt03a-1066 | Grobbendonk Troon | Kleine Nete | 587 | 1982 | 2018 | 36 |
| L01_491 | Oostvleteren | Poperingevaart | 64 | 1972 | 2018 | 46 |
| L01_492 | Reninge | Kemmelbeek | 88 | 1986 | 2018 | 32 |
| L01_496 | Merkem | Marktjevaart | 77 | 1986 | 2018 | 32 |
| L02_422 | Sint-Michiels | Kerkebeek | 93 | 1983 | 2018 | 35 |
| L02_425 | Oostkamp | Rivierbeek | 65 | 1983 | 2018 | 35 |
| L02_442 | Maldegem | Ede | 46 | 1984 | 2018 | 34 |
| L04_009 | Massemen | Molenbeek | 44 | 1987 | 2018 | 31 |
| L04_036 | Liezele | Molenbeek | 104 | 1975 | 2018 | 43 |
| L05_404 | Moorsele | Heulebeek | 73 | 1985 | 2018 | 33 |
| L06_342 | Nederzwalm | Zwalmbeek | 111 | 1972 | 2018 | 46 |
| L07_285 | Essene | Bellebeek | 90 | 1975 | 2018 | 43 |
| L07_286 | Sint-Katarina-Lombeek | Hunselbeek | 22 | 1983 | 2018 | 35 |
| L07_287 | Ternat | Steenvoordebeek | 26 | 1983 | 2018 | 35 |
| L07_289 | Viane | Mark | 123 | 1976 | 2018 | 42 |
| L08_112 | Heverlee | Voer | 49 | 1986 | 2018 | 32 |
| L08_115 | Heverlee | Molenbeek | 48 | 1986 | 2018 | 32 |
| L08_233 | Sint-Pieters-Leeuw | Zuunbeek | 65 | 1978 | 2016 | 38 |
| L09_136 | Hasselt | Demer | 270 | 1983 | 2018 | 35 |
| L09_138 | Bilzen | Demer | 116 | 1972 | 2018 | 46 |
| L09_145 | Ransberg | Velpe | 97 | 1975 | 2018 | 43 |
| L09_147 | Molenstede | Zwart Water | 79 | 1986 | 2018 | 32 |
| L09_156 | Rummen | Melsterbeek | 153 | 1983 | 2018 | 35 |
| L09_163 | Spalbeek | Herk | 274 | 1977 | 2018 | 41 |
| L11_022 | Overpelt | Dommel | 112 | 1971 | 2018 | 47 |
| L11_048 | Merksplas | Mark | 32 | 1983 | 2018 | 35 |
| L11_518 | Opoeteren | Bosbeek | 76 | 1985 | 2018 | 33 |
| LS06_347 | Etikhove | Molenbeek | 51 | 1972 | 2018 | 46 |
| LS09_165 | Wellen | Herk | 111 | 1972 | 2018 | 46 |

**Table 2. Drivers considered for this study**

Catchment specific *CAT*

| Topographic | Soil texture [% of total area] |
|---|---|
| Area [km$^2$], Slope [%] and Density [m/km$^2$] | Arenic, Clayic, Loamic, Loamic/Arenic, Loamic/Clayic, Loamic/Siltic, Siltic, Siltic/Clayic, Sitlic/Clayic/Loamic, Sitlic/Loamic |

Climate variability *CLIM* – weather types [% of time in a rolling window of 5 years]

| W; (NW, N); (NE, E, SE); (S, SW); A; C and U |
|---|

Land cover *LULC* [% of total area]

| Settlement, Agriculture, Grassland, Forest, Wetland and Other area |
|---|

**Table 3. Coefficients of the 26 terms in 9 predictors of the final model.**

| | | | | | |
|---|---|---|---|---|---|
| (Intercept) | -3.16 | A | 16.06 | Loam:Forest | 3.92 |
| Slope | 0.36 | Slope:Loam | 1.04 | Loam:Agriculture | 1.71 |
| Density | 0.05 | Slope:Silt | 0.75 | Loam:Settlement | 6.47 |
| Loam | -10.45 | Slope:Forest | -0.45 | Silt:Agriculture | -1.51 |
| Silt | -7.06 | Slope:Agriculture | -1.22 | Forest:A | -22.60 |
| Forest | 3.71 | Slope:Settlement | -0.85 | Agriculture:S_SW | -8.11 |
| Agriculture | 13.08 | Density:Loam | 66.84 | Agriculture:A | -17.77 |
| Settlement | -3.04 | Density:Silt | 73.81 | Settlement:A | 17.85 |
| S_SW | 11.13 | Density:Agriculture | -75.22 | | |

