# Peer review of "Climate or land cover variations: what is driving observed changes in river peak flows? A data-based attribution study"

_Hydrology and Earth System Sciences, 2018_

## Referee Comment (RC1) · Anonymous Referee #1 · 2 Aug 2018

I have read the manuscript entitled "Climate or land cover variations: what is driving observed changes in river peak flows? A data-based attribution study" with interest. The topic of the manuscript is suitable for the journal. Indeed, it has been widely referenced but the need to know the significance of drivers for floods in different areas still exists.

In this case, 29 catchments in Flanders were selected for the analysis of the influence of catchment characteristics, climate and land use variables on floods. In general, the objective of the paper is clear, "to investigate the (relative) importance of climate variability and land cover changes related to changes in river peak flows", and results

obtained are interesting. However, in my opinion, the authors should work further on the analysis and discussion of those results, trying to better explain the dependencies between drivers and floods, but specially the influence of interactions between drivers, that explain up to the 32% of the variability on river peak flows.

The abstract is clear and concise but it lacks some general conclusion.

Introduction is well structured, follows a clear central theme and mentions many references, that could be used to enrich the text extracting some information from them that could help esblishing the state of the art in the topic. Little is said in the introduction about the significance (in other studies) of one of the main drivers in this study: catchment characteristics, in my opinion some references should be included on this.

The case study is not sufficiently described. In my opinion, a general description of the area, considering average climate, geology, slopes, hydrology, vegetation should be included (are they spatially variable?), in order to have a general idea on the study area characteristics and the representativeness of the selected catchments.

- Table 1 includes the period and the number of years of discharge data for each gauging station. These information is repetitive and in my opinion not needed, as data used are those from 1992 to 2015 for all stations. Eliminating those columns may leave space enough for including data on fig. 3 (soil texture) in this table.

The methods section needs to be explained further as some questions are not clear enough: - Is daily discharge data an adequate time resolution to explore river peak flows in catchments smaller than 100 km2? Many of the catchments included in this study are quite small, so that discharge response, especially during peak flows, could be lower than the daily scale proposed; could the authors justify that the selected scale is adequate for the analysis of high flows?

- Using a figure/example/scheme could help understanding better the estimation of peak flow anomalies in section 3.2.

- In section 3.3. The consideration of characteristics other than climate and land use in the analysis is interesting; however, the authors should justify the inclusion of catchment characteristics on the analysis and the selection of the included characteristics. Why those and not others? The general description of the area may help on this, if the selected characteristics are the ones that show higher variability in the area...

- Nothing about soil is said in this section

- Some information is repeated in section 2 and section 3.3. The authors should decide where to include the completed information just once. For example: P3L7-9: "For land cover, the 30 classes from the ESA CCI Land Cover project (www.esa-landcover-cci.org) were regrouped into the 6 IPCC land categories, i.e. cropland, forest, grassland, wetland, settlement and other land...". P4L19-22: "Land cover and land cover changes have been described in the past through the ESA CCI project (www.esa-landcover-cci.org): . . .. The 22 land cover categories (or 30, when including 'level 2' or 'regional' labels) identified in this project are grouped into the six IPCC land categories, i.e. settlement, agriculture, grassland, forest, wetland and other area."

- In table 2 soil textures are included. However, nowhere in the text or in other figures and tables the authors talk about textures. They base their analysis in soil textures or in soil classes?? This should be clarified and corrected.

The first part of the Results section, that referred to steps followed "Prior to the first step of the model building process", should not be included in this section but in methodology as no results are explained here (P5L26-P6L4).

- Is it possible to consider at the same time variables that change spatially but not on time (catchment characteristics) and variables that change on space and time? How should results be considered? Catchment characteristics explain a high % of the variability in flood records, however, they are supposed to change only spatially, from one catchment to another, not for the same catchment from one year to the next. However, climate or landuse, explain less variability, but they change from one catchment to another and also in the same catchment on time. How should these be considered when analyzing results? Some discussion on this point would be interesting.

- P6L10: "The final model, with 26 terms in 9 predictors. . .". Which terms and predictors? The 9 predictors that in figure 5 are higher than the 50%? And terms? I would appreciate if the authors could specify a bit more.

- Figure 6 needs more explanation and discussion in the text. Which are the most problematic catchments? Can those worse results be related to some specific aspect/characteristic of the catchment?

- What about interactions between variables? How do they work? Which are the most significant? Does the same landuse change have same results in different climatic conditions? And for different chatchment characteristics? And what about climate variability? Has the same effect under forest or under agricultural land? What else can be extracted from figures 7 and 8?

- Figure 8 footnote should be corrected: "Increasing settlement area will, in most cases, lead to increased 5 peak flows". This is not what the figure shows but what the authors read from the figure. The figure shows boxplots shoing the results given by the model for all the catchments when increasing (or reducing?? See the text of P6L18) settlement percentage to reduce (or increase?) forest. . ..

- What does figure 8 really show? Contradictions are found in the text: P6L18: "1% of the total area from settlement to forest, grassland and agriculture, respectively" P7L17: "1% increase in urbanization could lead in some cases to a 5% increase in river peak flows"

- In this figure (8) it can be observed that changes in peak flows vary depending on which type of landuse is reduced to increase settlements. Could the authors say something about that? What do other authors say about it?

The discussion and conclusion section repeats 3 times that the model explains the

60% of the flood variability, but it does not discuss which could be the reasons why in some catchments the fitting or the consistency is not good. - In P6L29-30 the authors say "Since the explanatory variables all have a smooth variation over time, it is a priori almost impossible for any simple regression model to mimic these step changes". However, there are important changes in landuse around the year 2002.

- The comment on the time span used in the analysis (P6L32-P7L3) is not a conclusion and in my opinion, should not be included neither as a discussion.

- As the author say "Obviously, given the complexity of these environmental systems, the simple linear model will not be able to capture/describe all effects – indeed, it was seen that interaction effects between catchment characteristics, land cover and climate variability are equally important in explaining changes in river peak flows." In my opinion a deeper analysis of results and discussion on this part would notably improve the impact of the paper.

- P7L16-17: "The model also showed that, for most of the considered case studies, deforestation indeed leads to increased peak flows" where can this effect be seen? Deforestation? Or decreasing forest to increase settlement, agriculture or others?. "Moreover, 1% increase in urbanization could lead in some cases to a 5% increase in river peak flows". Can these results be analyzed a bit more? In which cases? Which characteristics have those catchments??

Other comments:

- Which is the resolution of the DTM mentioned in section 3.3? In P4L6 the authors say "The slope at every point in the catchment are calculated", which is the resolution of those points? (1x1; 5x5, meter?)

- P4L15. "W; (NW, N), (NE; E; SE), (S; SW); U; C; A, with N, E, S and W referring to wind directions". Please consider re-writing this sentence. Comma and semi-colon are arbitrary used. Parenthesis do not help understanding groups.

- In figure 3 the word fraction should be replaced or accompanied by classes not to create confusion with soil fractions (sand, silt and clay)

- Figure 3. Information included in this figure can be moved to table 1

- Reference list needs revision. For example: - "IPCC, 2014": review formatting. doi included corresponds to: IPCC, 2013: Climate Change 2013: The Physical Science Basis. Contribution of Working Group I to the Fifth Assessment Report of the Intergovernmental Panel on Climate Change [Stocker, T.F., D. Qin, G.-K. Plattner, M. Tignor, S.K. Allen, J. Boschung, A. Nauels, Y. Xia, V. Bex and P.M. Midgley (eds.)]. Cambridge University Press, Cambridge, United Kingdom and New York, NY, USA, 1535 pp, doi:10.1017/CBO9781107415324.

- "Blöschl, G., Ardoin-bardin, S., Bonell, M., Dorninger, M., Goodrich, D., Gutknecht, D., Matamoros, D., Merz, B., Shand, P. and Szolgay, J.: At what scales do climate variability and land cover change impact on flooding and low flows ?, Hydrol. Process., 1247(March), 1241–1247, doi:10.1002/hyp, 2007." doi is not complet. https://doi.org/10.1002/hyp.6669 and journal volume is 21.

- Mediero et al., 2015. Last author surname is not complete, lacks first letter.
* * *

---

## Referee Comment (RC2) · Anonymous Referee #2 · 3 Aug 2018

I decided to review this manuscript because the title was quite enticing. The manuscript was interesting overall, but it is simply too short the work done. Each of the figures regarding results should carry a paragraph, but most of them do only a sentence. I entirely agree with the comments by Reviewer 1, so thankfully I don't have to repeat here.

Reviewer 1 questions the variability by space of some variables and that by time of other variables. I understand that's why the authors used the panel data analysis, but it seems explained/discussed inadequately. I do not personally know the analysis method, and I still do not understand either.

[Figure]

I would like to point out that there are many paragraphs that are too short for a paragraph (e.g. line 12-13 on page 3). Please make them a complete set of thoughts. I also recommend that the authors separate discussion from conclusion. To me, the biggest problem with the manuscript is a lack of a general conclusion.

---

## Author Comment (AC1) · 10 Aug 2018

Dear reviewer

We thank you for your evaluation of our manuscript. We appreciate the constructive comments and suggestions you have made and would like to respond to them below.

Several comments deal with the lack of a discussion on the interaction effects. Hence, we investigated this further in detail and added a subsection in the results on these interaction effects. Also, we added some more discussion on the single driver effects. As such, the manuscript changed significantly, and we added a revised manuscript to

[Figure]

this reply.

GENERAL

I have read the manuscript entitled "Climate or land cover variations: what is driving observed changes in river peak flows? A data-based attribution study" with interest. The topic of the manuscript is suitable for the journal. Indeed, it has been widely referenced but the need to know the significance of drivers for floods in different areas still exists. In this case, 29 catchments in Flanders were selected for the analysis of the influence of catchment characteristics, climate and land use variables on floods. In general, the objective of the paper is clear, "to investigate the (relative) importance of climate variability and land cover changes related to changes in river peak flows", and results obtained are interesting. However, in my opinion, the authors should work further on the analysis and discussion of those results, trying to better explain the dependencies between drivers and floods, but specially the influence of interactions between drivers, that explain up to the 32% of the variability on river peak flows.

COMMENTS

Comment 1. The abstract is clear and concise but it lacks some general conclusion.

REPLY. We will add some more conclusions in the abstract, with respect to the interaction effects.

Comment 2. Introduction is well structured, follows a clear central theme and mentions many references, that could be used to enrich the text extracting some information from them that could help establishing the state of the art in the topic. Little is said in the introduction about the significance (in other studies) of one of the main drivers in this study: catchment characteristics, in my opinion some references should be included on this.

REPLY. We prefer to keep the introduction as it currently stands. With respect to the catchment characteristics: based on some of the other comments, the results section

will significantly change in the revised manuscript. And the effects of single drivers will be discussed more in depth, where our results will be compared with literature.

Comment 3. The case study is not sufficiently described. In my opinion, a general description of the area, considering average climate, geology, slopes, hydrology, vegetation should be included (are they spatially variable?), in order to have a general idea on the study area characteristics and the representativeness of the selected catchments.

REPLY. We will add below paragraphs under the section Case study: For this case study, 29 catchments are selected, evenly spread across Flanders, the Northern part of Belgium (Figure 1). Flanders, with 6.4 million inhabitants, covers around 13,500 km2. The coastal area in the North-West of the region mainly consists of sand dunes and clayey alluvial soils in the polders. The central area, ranging between 0 and 10 mTAW, mainly consists of loamic soils. The North-Eastern part, known as the Campine region, has sandy soils at altitudes around 30 mTAW. The Southern part with silty soils has low hills op to 150 mTAW. The maximum height is 288 mTAW in the South East. The DTM in Figure 1 was taken from "Digitaal Hoogtemodel Vlaanderen" (https://overheid.vlaanderen.be/producten-diensten/digitaal-hoogtemodel-dhmv). Soil texture is available from www.dov.vlaanderen.be. Flanders has a maritime climate (Cfb, according to the Koppen climate classification), with average temperatures of 3 °C and 18 °C in January and July, respectively. There is a small gradient present with lower temperatures in the South-East (annual average of 10 °C) towards higher temperatures in the North-West (annual average of 11 °C) (data between 1981-2010); the average temperatures in Flanders, further, has been rising over the past 30 years with 1 – 1.5 °C. Average evapotranspiration was 540 mm/year in 1980 and rose to 625 mm/year in 2010. Yearly precipitation varies between 600 mm/year to 1000 mm/year, with little variation throughout the year, and little spatial differences (Brouwers et al., 2015).

Comment 4. Table 1 includes the period and the number of years of discharge data for each gauging station. These information is repetitive and in my opinion not needed, as

data used are those from 1992 to 2015 for all stations. Eliminating those columns may leave space enough for including data on fig. 3 (soil texture) in this table.

REPLY. We would like to keep table 1 as it currently stands. To estimate peak flow anomalies, we use all available data (e.g. for Grobbendonk Troon, this is 1982 – 2018). Then, the variation is explained based on the selected predictors, and due to the availability of these predictors, the period is reduced. Further, see also the comment lower related to Fig. 3 – we wish to keep Fig. 3 as is, in order to make a visual comparison between the catchments possible (wrt soil texture).

The methods section needs to be explained further as some questions are not clear enough: Comment 5. Is daily discharge data an adequate time resolution to explore river peak flows in catchments smaller than 100 km2? Many of the catchments included in this study are quite small, so that discharge response, especially during peak flows, could be lower than the daily scale proposed; could the authors justify that the selected scale is adequate for the analysis of high flows?

REPLY. Please note that, unfortunately, in the manuscript it was erroneously noted as daily discharge data (P3L16 and P3L21). However, the analysis has been carried out based on hourly data. This will be corrected in the revised manuscript to state hourly discharge data was used. The hourly discharge data is freely available through https://www.waterinfo.be/default.aspx?path=NL/Rapporten/Downloaden, under section 1A Waterlevel/riverflow select: Discharge (Hourly), and you can verify this hourly data is available.

Comment 6. Using a figure/example/scheme could help understanding better the estimation of peak flow anomalies in section 3.2.

REPLY. In the revised manuscript, we will add a figure under Section 3.2 and replace the text in order to better explain the estimation of peak flow anomalies. See the revised manuscript in supplement to this reply.

Comment 7. In section 3.3. The consideration of characteristics other than climate and land use in the analysis is interesting; however, the authors should justify the inclusion of catchment characteristics on the analysis and the selection of the included characteristics. Why those and not others? The general description of the area may help on this, if the selected characteristics are the ones that show higher variability in the area...

REPLY. Soil texture taken into account as there are significant gradients in Flanders and thus differences amongst the various catchments: e.g. L01_491 has mainly a loamic soil texture, whereas L07_286 is mainly siltic, etc. Slopes should definitely be taken into account, as this has a known/obvious impact on rainfall runoff. Similar for the river density (ratio of river length over catchment area). Catchment area is often linked to peak flow sensitivity, and thus was initially taken into account for this study. However, later in the study, this variable is discarded, based on statistical considerations. See also the start of the discussion (P6L21). Note that these characteristics also come back in the concepts of the hydrological model WetSpa when assigning runoff coefficients.

Comment 8. Nothing about soil is said in this section

REPLY. Soil texture is mentioned as possible drivers, P4L9. We don't see what you are missing here?

Comment 9. Some information is repeated in section 2 and section 3.3. The authors should decide where to include the completed information just once. For example: P3L7-9: "For land cover, the 30 classes from the ESA CCI Land Cover project (www.esa-landcovercci.org) were regrouped into the 6 IPCC land categories, i.e. cropland, forest, grassland, wetland, settlement and other land...". P4L19-22: "Land cover and land cover changes have been described in the past through the ESA CCI project (www.esalandcover-cci.org): . . .. The 22 land cover categories (or 30, when including 'level 2' or 'regional' labels) identified in this project are grouped into the six IPCC land categories, i.e. settlement, agriculture, grassland, forest, wetland and other area."

REPLY. Indeed, this was repeated in both sections. We would adjust the text in Section 3.3 to: "Six IPCC land categories (settlement, agriculture, grassland, forest, wetland and other area) are taken into consideration as possible drivers for this study.."

Comment 10. In table 2 soil textures are included. However, nowhere in the text or in other figures and tables the authors talk about textures. They base their analysis in soil textures or in soil classes?? This should be clarified and corrected.

REPLY. Analysis is based on soil textures – not soil classes. We will scan the manuscript and clarify/correct where needed.

Comment 11. The first part of the Results section, that referred to steps followed "Prior to the first step of the model building process", should not be included in this section but in methodology as no results are explained here (P5L26-P6L4).

REPLY. OK. This will move to the relevant paragraphs of the methods section in the revised manuscript.

Comment 12. Is it possible to consider at the same time variables that change spatially but not on time (catchment characteristics) and variables that change on space and time? How should results be considered? Catchment characteristics explain a high % of the variability in flood records, however, they are supposed to change only spatially, from one catchment to another, not for the same catchment from one year to the next. However, climate or landuse, explain less variability, but they change from one catchment to another and also in the same catchment on time. How should these be considered when analyzing results? Some discussion on this point would be interesting.

REPLY. Yes, temporal and spatial data can be considered simultaneously – through panel data analysis.

Catchment characteristics only show a spatial variation and no temporal variation. And, indeed, they explain a high % of the variability in flood records. Meaning that the

flood responses is strongly catchment specific, and in a lesser degree depending on fluctuations of the climate, and land use changes. For the revised manuscript, we will change the first paragraphs of 3.4.1 (P4L26-29) to: "A model is built with the techniques and ideas of panel analysis, which is widely used in social sciences, epidemiology, and econometrics where two dimensional data is analysed. Typically, in those sectors data is collected over time and over the same individuals. Here, the two dimensions are space and time – input data can show only a temporal variation (climate data), only a spatial variation (soil texture), or a combination of both (LULC). Note that, typically, climate data does show a spatial variation as well. However, we assume the area of Flanders to be homogeneous with respect to the considered climate data."

And, we would add in the conclusion (P7L13) "[. . .] topography and soil texture. The high importance of these time-invariant factors (topography and soil texture) indicate flood response in Flanders is highly catchment specific, and to a lesser degree depending on fluctuations of the climate, and land use changes."

Comment 13. P6L10: "The final model, with 26 terms in 9 predictors. . .". Which terms and predictors? The 9 predictors that in figure 5 are higher than the 50%? And terms? I would appreciate if the authors could specify a bit more.

REPLY. The predictors: indeed the 9 predictors in Figure 5 that are higher than the 50%. We believe this should be clear by P6L5-9.

We will include the coefficients of the final model in the revised manuscript (see table 3 in the revised manuscript in supplement to this reply). One should, however, be careful when interpreting these coefficients. E.g. the coefficient of Settlement in the final model is equal to -3.04. At first sight, an increase of settlement would thus correspond with a decrease of peak flow anomaly. However, the interpretation of the coefficients is more complex:

An increased Settlement also impacts the interaction effects, and the coefficient becomes: $(-3.04 - 0.85*Slope + 6.47*Loam + 17.85*Settlement)$;

An increased Settlement means that Agriculture (13.08) and/or Forest (3.71) decrease – and there again, the interaction effects of Agriculture and Forest come into play. Thus, because of the dependency between variables, the interpretation of the resulting model coefficients ïĄć is not straightforward. Therefore, we did not include this table initially. However, based on this comment, we will include it in the revised manuscript.

Comment 14. Figure 6 needs more explanation and discussion in the text. Which are the most problematic catchments? Can those worse results be related to some specific aspect/characteristic of the catchment?

REPLY. Observed peak flow anomalies in catchments L07_289 (Mark at Viaene) and L08_233 (Zuunbeek at Sint-Pieters-Leeuw) have a bad correspondence with their modelled results. The Mark catchment has a long history of flooding – as from the 2000s, the local authorities have installed several mitigation measures (hydraulic structures, retention basins etc.), effectively decreasing the flood risk. This is visible in the observed peak flow anomaly, however, the regression model used in this study cannot capture such management changes. Further, for the Zuunbeek catchment at Sint-Pieters-Leeuw, increased peak flow anomalies are observed as from the middle of the period. This is due to the extreme flood season in the winter of 2001-2002 where 7 events were observed with peak discharges exceeding 6 m3/s, corresponding to empirical return periods larger than 1 year, based on data between 1978 and 2016. We will add the above paragraph in the Results section of the revised manuscript.

Comment 15. What about interactions between variables? How do they work? Which are the most significant? Does the same landuse change have same results in different climatic conditions? And for different catchment characteristics? And what about climate variability? Has the same effect under forest or under agricultural land? What else can be extracted from figures 7 and 8?

REPLY. A section on interaction effects will be added in the revised manuscript.

Comment 16. Figure 8 footnote should be corrected: "Increasing settlement area will,

in most cases, lead to increased 5 peak flows". This is not what the figure shows but what the authors read from the figure. The figure shows boxplots shoing the results given by the model for all the catchments when increasing (or reducing?? See the text of P6L18) settlement percentage to reduce (or increase?) forest. . ..

REPLY. We will change the caption in the revised manuscript to: "Figure 8. Peak flow changes by increasing settlement area trough decreasing forest, grassland or agriculture."

Comment 17. What does figure 8 really show? Contradictions are found in the text: P6L18: "1% of the total area from settlement to forest, grassland and agriculture, respectively" P7L17: "1% increase in urbanization could lead in some cases to a 5% increase in river peak flows"

REPLY. We would replace the word 'urbanisation' on P7 to 'settlement'. We further don't see the contradiction? Ref. also comment higher on the caption of this figure.

Comment 18. In this figure (8) it can be observed that changes in peak flows vary depending on which type of landuse is reduced to increase settlements. Could the authors say something about that? What do other authors say about it?

REPLY. Indeed, we specifically quantified the changes in peak flows for increased urban area depending on the type of LULC that is reduced. In literature, we did not find similar studies where the independent effects were quantified. Most of the references look at the total picture by comparing the situation for two distinct periods in time and, as such, observing/modelling the integral response of a catchment due to e.g. the simultaneous decreases in forested area and agricultural area in favor of urban area.

Comment 19. The discussion and conclusion section repeats 3 times that the model explains the 60% of the flood variability, but it does not discuss which could be the reasons why in some catchments the fitting or the consistency is not good. In P6L29-30 the authors say "Since the explanatory variables all have a smooth variation over

time, it is a priori almost impossible for any simple regression model to mimic these step changes". However, there are important changes in landuse around the year 2002.

REPLY. Ref. comment 14, and reply on this comment, higher. We now looked deeper into the reasons why we see a bad fit for some catchments. For the revised manuscript, we will further take out the comment on the explanatory variables and their smooth variation over time.

Comment 20. The comment on the time span used in the analysis (P6L32-P7L3) is not a conclusion and in my opinion, should not be included neither as a discussion.

REPLY. Okay, we will delete this paragraph in the revised manuscript.

Comment 21. As the author say "Obviously, given the complexity of these environmental systems,the simple linear model will not be able to capture/describe all effects – indeed, it was seen that interaction effects between catchment characteristics, land cover and climate variability are equally important in explaining changes in river peak flows." In my opinion a deeper analysis of results and discussion on this part would notably improve the impact of the paper.

REPLY. A section on interaction effects will be added in the revised manuscript.

Comment 22. P7L16-17: "The model also showed that, for most of the considered case studies, deforestation indeed leads to increased peak flows" where can this effect be seen? Deforestation? Or decreasing forest to increase settlement, agriculture or others?. "Moreover, 1% increase in urbanization could lead in some cases to a 5% increase in river peak flows". Can these results be analyzed a bit more? In which cases? Which characteristics have those catchments??

REPLY. Correct, the use of the term "deforestation" might be out of place here, as, indeed, it is rather decreasing forest to increase settlement and others. Will replace this sentence to: "The model also showed that, for most of the considered case studies, a decrease in forested area to increase settlement area indeed leads to increased peak

flows."

The catchment with the strongest influence are flat catchments with a high loamic content. In this case: L01_491, L01_492, L01_496 and L05_404 have the highest impacts. This confirms the results from the interaction effects, which will be described in a separate subsection of the revised manuscript.

Other comments: Comment 23. Which is the resolution of the DTM mentioned in section 3.3? In P4L6 the authors say "The slope at every point in the catchment are calculated", which is the resolution of those points? (1x1; 5x5, meter?)

REPLY. Resolution of the DTM is 100x100m. Will add this on P4L5 of the original manuscript.

Comment 24. P4L15. "W; (NW, N), (NE; E; SE), (S; SW); U; C; A, with N, E, S and W referring to wind directions". Please consider re-writing this sentence. Comma and semi-colon are arbitrary used. Parenthesis do not help understanding groups.

REPLY. Correct, comma and semi-colon were unfortunately arbitrarily used. Groups are separated by parentheses and semi-colons. Weather types within each group are separated by commas. We hope the following is more clear: "The remaining weather types are: W; (NW, N); (NE, E, SE); (S, SW); U; C; A, with N, E, S and W referring to wind directions, C and A to cyclonic and anticyclonic atmospheric patterns, respectively, and U to an unclassified weather type."

Comment 25. In figure 3 the word fraction should be replaced or accompanied by classes not to create confusion with soil fractions (sand, silt and clay)

REPLY. Okay, will be replaced by "Figure 3. Relative areas of soil texture classes (arenic, loamic and siltic) for the selected catchments. Data from: www.dov.vlaanderen.be" in the revised manuscript.

Comment 26. Figure 3. Information included in this figure can be moved to table 1.

REPLY. Correct, this could be included in Table 1. However, we specifically opted for this graphical representation. In this way, the reader can more easily see what the dominant soil class is for that particular catchment. The order of the catchments is the same as in Figure 2 and Figure 6, which makes a comparison between these figures easier. Also, with the current graphical representation, it is easier to compare two (or more) catchments among each other – in table format, this is a bit more difficult.

Reference list needs revision. For example:

Comment 27. "IPCC, 2014": review formatting. Doi included corresponds to: IPCC, 2013: Climate Change 2013: The Physical Science Basis. Contribution of Working Group I to the Fifth Assessment Report of the Intergovernmental Panel on Climate Change [Stocker, T.F., D. Qin, G.-K. Plattner, M. Tignor, S.K. Allen, J. Boschung, A. Nauels, Y. Xia, V. Bex and P.M. Midgley (eds.)]. Cambridge University Press, Cambridge, United Kingdom and New York, NY, USA, 1535pp, doi:10.1017/CBO9781107415324.

REPLY. Indeed, doi is referring to the wrong document. In the revised manuscript, this reference will be updated and formatted according to guidelines mentioned on http://www.ipcc.ch/report/ar5/syr/.

Comment 28. "Blöschl, G., Ardoin-bardin, S., Bonell, M., Dorninger, M., Goodrich, D., Gutknecht, D., Matamoros, D., Merz, B., Shand, P. and Szolgay, J.: At what scales do climate variability and land cover change impact on flooding and low flows ?, Hydrol. Process., 1247(March), 1241–1247, doi:10.1002/hyp, 2007." doi is not complet. https://doi.org/10.1002/hyp.6669 and journal volume is 21.

REPLY. This will be corrected in the revised manuscript.

Comment 29. Mediero et al., 2015. Last author surname is not complete, lacks first letter.

REPLY. This will be corrected in the revised manuscript.

Please also note the supplement to this comment:
https://www.hydrol-earth-syst-sci-discuss.net/hess-2018-385/hess-2018-385-AC1-supplement.pdf

———————————————————

[Figure]

**Supplement:**

[revised manuscript text omitted]

---

## Author Comment (AC2) · 10 Aug 2018

Dear reviewer

Thank you for reading our manuscript and providing us feedback. Mainly based on the comments raised by Reviewer 1, we significantly expanded the manuscript. A revised manuscript can be found in the supplement to the reply of Reviewer 1. Also, as you agreed on the comments of Reviewer 1, we would invite you to also read our reply to these comments.
* * *
385, 2018.

---

## Referee Comment (RC3) · Anonymous Referee #3 · 11 Oct 2018

In general, this manuscript covers an interesting topic and makes some conclusions leading to new thinkings regarding the impacts of climate change and land use/cover change on water resources. However, to the manuscript itself, I made a few suggestions and comments as a potential reader.

1. Page 1 Line 7 - "increased urbanization": Not only urbanization but also some other land use/cover categories can have a great impact on the hydrological cycle. This word is recommended to be revised.

2. Page 1 Line 19 - "...it is very likely that further changes will...": You have to give a citation here or tell readers why you can make this conclusion.

[Figure]

3. Page 2 Line 1 - "...knowledge on some driver-effect mechanisms is still limited (Blöschlet al., 2007; Merz et al., 2012)": When you make the conclusion of the limitation in this study, recent articles should be cited in order to inform the reader that the conclusion is not out of date.

4. Page 2 Line 15 - "...mainly because of the heterogeneity in catchments globally and the scale of the river basin/catchment considered": What kind of heterogeneity catchments have? (heterogeneity of land use/cover change or heterogeneity of hydrological responses) Moreover, citations should be given to the conclusion here if this result is not a part of your research.

5. Page 2 Line 30 - Use Section instead of Sect.

6. Page 2 Line 32 - As you are talking about the study area. "Study area and data" is suggested to be used for this section.

7. Page 2 Line 35 - Give the full explanation of mTAW.

8. Page 3 Line 9 - You have to make the consistent on the numeric format. Please be advised to use 29 instead of Twenty-nine.

9. Page 3 Line 14&15 - " For soil texture, taken from www.dov.vlaanderen.be, 3 domina...": Minor grammar suggestion. You can use "soil texture data is obtained from..."

10. Page 3 Line 21 - "The aim of the study is to find the (main) drivers ...": You have to make a decision whether including the word in the bracket or not.

11. Page 4 Line 6 - "...based on the DTM, the slope at every point in the catchment...": You do not have to say at every point; this sentence is suggested to be revised to "the slope in the catchment".

12. Page 5 Line 14 - Equation 2: What is captalized T?

13. Page 6 Line 12 - "...the model did not improve (not shown)...": The model did not improve what?

14. Page 7 Line 11 - "...as seen in Figure 9:": Figure 9 or Figure 10?

15. Page 13 - Figure 2: You don't have to give the data source here.

I read the revised manuscript which the author provided in the first referee's comment place. Although there are many places have been strengthened, I think this manuscript needs to be improved in grammar checking and make clear meanings in some sentences as well.

---

## Author Comment (AC3) · 12 Oct 2018

Dear reviewer Thanks for your evaluation of our manuscript. We appreciate your comments and suggestions and would like to respond to them below. See also the supplement to this reply for a draft revised manuscript, incorporating the suggested changes.

COMMENT 1. Page 1 Line 7 - "increased urbanization": Not only urbanization but also some other land use/cover categories can have a great impact on the hydrological cycle. This word is recommended to be revised.

REPLY. OK. This will be changed to "land use/land cover changes" in the revised

manuscript.

COMMENT 2. Page 1 Line 19 - "...it is very likely that further changes will...": You have to give a citation here or tell readers why you can make this conclusion.

REPLY. The reason why we make this conclusion, is expressed further in that paragraph: climate change projections from IPCC (2014) and impact analysis from Tabari et al. (2015), and UN (2018), Poelmans (2010) and Ruimte Vlaanderen (2017) talk about future changes in the built environment.j

COMMENT 3. Page 2 Line 1 - "...knowledge on some driver-effect mechanisms is still limited (Blöschl et al., 2007; Merz et al., 2012)": When you make the conclusion of the limitation in this study, recent articles should be cited in order to inform the reader that the conclusion is not out of date.

REPLY. Okay, we will add more recent references in the revised manuscript: Van Loon et al (2016) and Dey and Mishra (2017).

COMMENT 4. Page 2 Line 15 - "...mainly because of the heterogeneity in catchments globally and the scale of the river basin/catchment considered": What kind of heterogeneity catchments have? (heterogeneity of land use/cover change or heterogeneity of hydrological responses) Moreover, citations should be given to the conclusion here if this result is not a part of your research.

REPLY. Here, we mean the heterogeneity of hydrological responses of the catchments, partially originating in the heterogeneity of lulc changes. We will clarify this in the revised manuscript, and give the reference to Zhang et al (2017) as an example.

COMMENT 5. Page 2 Line 30 - Use Section instead of Sect.

REPLY. OK. This will be changed in the revised manuscript.

COMMENT 6. Page 2 Line 32 - As you are talking about the study area. "Study area and data" is suggested to be used for this section.

REPLY. OK. This will be changed in the revised manuscript.

COMMENT 7. Page 2 Line 35 - Give the full explanation of mTAW.

REPLY. mTAW is the local height datum, equal to the height, in meters, above local mean seal level. This will be added in the revised manuscript.

COMMENT 8. Page 3 Line 9 - You have to make the consistent on the numeric format. Please be advised to use 29 instead of Twenty-nine.

REPLY. According to the "manuscript preparation guidelines for authors", numbers are spelled out when they begin a sentence. Therefore, we use "Twenty-nine" here. Therefore, we suggest to leave this as it currently stands.

COMMENT 9. Page 3 Line 14&15 - " For soil texture, taken from www.dov.vlaanderen.be, 3 domina...": Minor grammar suggestion. You can use "soil texture data is obtained from..."

REPLY. OK. We will adjust the revised manuscript accordingly.

COMMENT 10. Page 3 Line 21 - "The aim of the study is to find the (main) drivers ...": You have to make a decision whether including the word in the bracket or not.

REPLY. OK. The brackets will be left out in the revised manuscript.

COMMENT 11. Page 4 Line 6 - "...based on the DTM, the slope at every point in the catchment...": You do not have to say at every point; this sentence is suggested to be revised to "the slope in the catchment".

REPLY. OK. This will be left out in the revised manuscript.

COMMENT 12. Page 5 Line 14 - Equation 2: What is captalized T?

REPLY. Capitalized T is the transpose of the vector/matrix. This will be added in the revised manuscript.

COMMENT 13. Page 6 Line 12 - "...the model did not improve (not shown)...": The

model did not improve what?

REPLY. We will change this to " . . . model performance did not improve . . .".

COMMENT 14. Page 7 Line 11 - "...as seen in Figure 9:": Figure 9 or Figure 10?

REPLY. Figure 9. The idea is here that we link conclusions from Figure 9 with the Wetspa model (Figure 10). We will change this sentence to: "As such, findings with respect to the potential runoff coefficient from Wetspa can be related with the conclusions based on Figure 9".

COMMENT 15. Page 13 - Figure 2: You don't have to give the data source here.

REPLY. OK. We will delete the reference in the revised manscript.

Please also note the supplement to this comment:
https://www.hydrol-earth-syst-sci-discuss.net/hess-2018-385/hess-2018-385-AC3-supplement.pdf

[Figure]

**Supplement:**

[revised manuscript text omitted]

---

## Referee Report (RR1)

**The First Paragraph**

It appears that the author undertook a substantial revision. The authors emphasize that both individual drivers and interaction terms are important in explaining the observed changes in river peak flows.

The main problem I had with the first submission was a lack of details and general conclusions. The problem was largely addressed in the revision, but I am not quite satisfied yet.

First, the abstract ends with a sentence saying interaction terms explain up to 32%. It appears to end again without a general conclusion. What is the implication of the interaction terms explaining 32%? Did the authors want to say interaction terms must be considered in such studies or something else? Then please say explicitly. After reading the abstract, my eyebrows went up and I thought "is this the end?"

Second, there are still numerous areas that need minor changes. Hereby I list them from the beginning:
- The authors use 'e.g.' too often. Most of them are quite annoying and 'for example' should be used instead.
- P1: Words 'our' appear a couple of times. I recommend removing/replacing them because the meaning is vague.
- P1L33: I am not sure why the colon is used
- P2L7: "most studies hypothesized that deforestation…" I wonder whether the studies 'hypothesized' or actually 'found' that deforestation cause increased surface runoff. If they merely hypothesized, then what happened to the hypotheses?
- P2L26: "Section" and "Sect." are used mixed in the document
- Section 2: the authors added a section about data availability at the end. Therefore, all the URLs in this section seem unnecessary (and are annoying)
- P3L1: "Koppen" has an Umlaut
- P3L4: "1 – 1.5" means one minus one point five. Should be "1-1.5"
- P3L24: "proposed by (Willems, 2009)" Please correct the citation format. There are other such cases in the document
- P4L15: the authors mix commas and semicolons in parentheses
- P5L20: "on (100 times) 20 random" I don't understand what the authors meant to say
- P6L2: "17.85*Sediment" I wonder why sediment is here. In addition, in this sentence, what did you mean by "the coefficient becomes –3.04 – 0.85*Slope…"? Did you put 1 for Sediment? Then again why Sediment is left in the last term?
- P6L5: I don't see how Figure 7 shows that the model explains 60% of the changes. Seeing the figure again, I still don't get it.
- P7L2: I don't understand "we do make any statements…"
- P7L12-13: 10%+6%+6% = 22%. Does it mean that there is additional 10% you did not further explain?
- P7L18-19: authors say "with increasing slope,…" but Figure 12(a) shows decreasing slope from left to right. I recommend plotting the graph with increasing slope from left to right, consistent with (b) and (c).

- P8L28-29: "The model also showed that, for most of the considered case studies,…" What did you mean by the considered case studies? Are they previous studies? If so, how are they relevant here?
- P8L14-16: I don't think these sentences are appropriate for the conclusion

---

## Author Response (AR2)

Decision: reconsider after major revisions

Dear Jan De Niel and Patrick Willems,

Thank you for revising the manuscript and responding to the referees' reports. Although I find this version of the paper much improved in comparison to the initial submission, I believe the manuscript can, and need to, be improved before I can consider it for publication. Please note that most of my comments below and the comments from the two referees mainly refer to the presentation quality.

Thanks for the reconsideration of our manuscript, and your suggestions for improvement (in addition to those of the referees).

Text editing are needed. There are typos and grammatical errors that should be corrected (some mentioned by the reviewers).

We went through the manuscript again, and hope most of the typos and grammatical errors are now corrected.

Introduction section. I agree with the comments made by Reviewer #1 - instead of giving a long list of references please elaborate and explicitly refer to the contribution of the papers mention to this study. In general, the introduction section needs to explain more in depth the climate and land cover drivers.

We expanded on the literature review in paragraph 3 and 4 in the Introduction.

There is a disproportion between the length of the text (8 pages) and the number of figures (13) and tables (3). I think 4-6 figures and 1-2 tables should be sufficient to present your study properly. Please consider merging some of the figures (for example, 12 and 13) and presenting some as Supplementary Information (e.g. figure 3 and table 1).

We combined Figures 12 and 13; and moved some of the figures and tables to Supplementary Material. (see also last comment of $2^{nd}$ reviewer)

Figure 4. Please added in the figure caption text to explain subplots (a) to (e).

OK. We added more explanation in the caption.

Please consider having separate sections to present the results and for the discussion.

We prefer to keep the results and discussion in one section, as we feel this is more appropriate for this manuscript.

Catchments. Sometimes you are using numbers and sometimes names when referring to the catchments. Please be consist. In addition, I suggest simplifying the labels of the catchments, e.g. use simple IDs from 1 to 29.

OK. We changed the IDs as suggested (1 to 29). However, for reference purposes, we suggest to keep the original labels in Table 1.

I invite you to upload a revised manuscript, incorporating the proposed changes and additions, and making any other modifications where you see fit ('major revision'). The revised manuscript will be sent for the referees for a third round of revision.

I look forward to receiving the revised manuscript.

Sincerely,

Nadav Peleg

Decision: reconsidered after major revisions.

I have read the manuscript entitled "Climate or land cover variations: what is driving observed changes in river peak flows? A data-based attribution study" for the second time. I still think the topic of the manuscript is suitable for the journal, and o great interest. The authors tried to address most of the comments of this reviewer (however, some comments were not considered or worked enough), nevertheless, in my opinion the text should be edited by a native English Editor for improving readability an understandability, before being published, as sometimes is difficult to follow the text.

Addittionally I still have some concern on the text that I list below:

**1. Introduction**

I still think there are a lot of references in the introduction and little content on them. The authors say they prefer to leave it as is, however, for me it does not make sense to refer to many other paper and not mention some ideas or conclusions of those researchers (paragraphs 3 and 4 of the introduction). In fact, the authors do comment on other authors findings on the next paragraph (number 5). In my opinion, this should be revised before publication.

Expanded on the literature review in paragraphs 3 and 4 in the introduction.

P2 L10-13: Do the references follow any order?

This is be alphabetical, based on the last name of the first author.

P1 L8: the objectives should go at the end of the introduction section, so that they can be deleted from this paragraph.

If this is referring to the sentence "The relative importance of both drivers, however, is still uncertain and interaction effects between both drivers are not yet well understood" from the abstract, we prefer to leave this sentence in the abstract. In our opinion, the objective should be mentioned (briefly) in the abstract.

P2 L14: "do not aim to attribute the changes to the specific type of changes that occur". Please clarify this sentence.

The clarification of this sentence comes after the colon. Changed this from: "e.g. an increase in settlement at the expense of agricultural land" to "for example, the isolated effect of an increase in settlement area at the expense of agricultural land would not be quantified".

P2 L15: "mainly because of the heterogeneity of hydrological responses. What do you refer to?" examples?

Included the example of Zhang et al. (2017): "for example, Zhang et al. (2017) found that small mixed forest-dominated watersheds and large snow-dominated watershed are more hydrologically resilient to forest cover change with respect to annual flows."

**2. Study area and data**

P2 L37-38. "The Southern part with silty soils has low hills op to 150 mTAW. The maximum height is 288 mTAW in the South East." If hills are up to 150 m how can maximum height be 288 m?, please clarify this sentence. Change op by up . I do not think it is necessary to repeat all the time TAW, it makes reading more difficult.

150mTAW vs. 288mTAW: 150 m is in the central Southern part; 288m is in the South East. Rephrased: "The central Southern part with silty soils has low hills up to 150 m. In the South East, the maximum height is 288 m."

Changed "op" to "up".

Removed TAW, except for the first time ("0 and 10 mTAW, with mTAW […]").

P3 L3-8. In this text the authors give general average data for the study area. However, they give specific data of ETR for years 1980 and 2010, a general average data, would be more appropriate to be compared with the rest of data given here.

This rise between 540 mm/year to 625 mm/year was an (almost) linear increase. Added this in the manuscript: "Average evapotranspiration was 540 mm/year in 1980 and increased almost linearly to 625 mm/year in 2010"

P3 L9: "Twinty" should be twenty.

Corrected this in the revised manuscript.

**3. Methods**

About table 1 and periods used to estimate peak-flow anomalies. How can influence the use of different periods in the estimation of those anomalies?. Shouldn't the same period for all catchments be used? Natural climate variability can be high from one year to other and it can strongly influence hydrological response. The authors should address this question and justify the use of different periods.

Indeed, different periods were used for estimation of the peak-flow anomalies: the longest period is 1971-2018 (47 years); the shortest period is 1987-2018 (31 years). There is a trade-off between (a) using the same period for all catchments, and thus limiting the data for all catchments to 1987-2018; and (b) making use of all available data, working with different periods for the various catchments, and keeping all available data. Both would be valid options to apply our methodology; we chose for the second option. We want to use the available data to the maximum extent possible to obtain the most reliable signal of peak-flow anomalies over time per catchment. Note that the 'reference peak-flow' per catchments (i.e. peak-flow anomaly equal to 1) is obtained through averaging over the whole period. Because of this averaging process, the anomalies would only undergo a small (positive or negative) shift when working with a slightly shorter period. This, in the end, would be captured by the estimation of the regression model. Therefore, we chose to use all available data.

Figure 4 was added in order to better explain methodology. However, some more explanation on the figure is needed in the footnote in order to understand it. What are figures a) b) c)….? This could be a general comment for figures and tables, as their foots are sometimes scarce on content and hardly explain what the figure or table means.

Changed the caption of Figure 4. See also the comment of the Editor.

On a previous comment: "In section 3.3. The consideration of characteristics other than climate and land use in the analysis is interesting; however, the authors should justify the inclusion of catchment characteristics on the analysis and the selection of the included characteristics. Why those and not others? The general description of the area may help on this, if the selected characteristics are the ones that show higher variability in the area..." The authors response is: "Soil texture taken into account as there are significant gradients in Flanders and thus differences amongst the various catchments: e.g. L01_491 has mainly a loamic soil texture, whereas L07_286 is mainly siltic, etc. Slopes should definitely be taken into account, as this has a known/obvious impact on rainfall runoff. Similar for the river density (ratio of river length over catchment area). Catchment area is often linked to peak flow sensitivity, and thus was initially taken into account for this study. However, later in the study, this variable is discarded, based on statistical considerations. See also the start of the discussion (P6L21). Note that these characteristics also come back in the concepts of the hydrological model WetSpa when assigning runoff coefficients." I agree with the answer, however, I don't see any of it in the new text of the manuscript.

The above answer to the previous comment is indirectly captured in the manuscript. In section 2 we state that some parts are clayey, loamic or sandy; average heights and presence of low hills are also mentioned. This indicates the variability of the selected drivers which are more explicitly introduced in Section 3.3.

As the variability is mentioned in Section 2, and in order to keep the manuscript concise, we prefer not to expand too much on this in Section 3.3.

P5 L3: "However, we assume the area of Flanders to be homogeneous with respect to the considered climate data" What do you mean? only one climatic series is considered for the whole area? Or, are you referring to weather types? Please clarify in the text, How can this affect results? I mean, considering it as homogenous?

Changed in the manuscript to: "However, in this study, climate variability is described through weather types and we assume the area of Flanders to be homogeneous with respect to these weather types".

P6 L34-37: "Further, for the Zuunbeek catchment at SintPieters-Leeuw, increased peak flow anomalies are observed as from the middle of the period. This is due to the extreme flood season in the winter of 2001-2002 where 7 events were observed with peak discharges exceeding 6 m3/s, corresponding to empirical return periods larger than 1 year, based on data between 1978 and 2016. We will add the above paragraph in the Results section of the revised manuscript." This makes me think about two previous questions. Is the period important?, can, in this case, different periods be used for anomaly estimation? Can the climate really be considered homogeneous in the study area?

With respect to the period: for the particular case of the Zuunbeek catchment, the step change would always be visible as long as the winter of 2001-2002 is in the period. This winter of 2001-2002 for the Zuunbeek is an very exceptional season (7 events with empirical return period > 1 year), and it is not within the scope of the regression model to explain this kind of extreme events/seasons. Further, as mentioned as a response to a previous question, if this winter would be left out for the estimation of peak-flow anomalies, the reference peak-flow (anomaly equal to 1) would only slightly shift downwards; the difference will be captured by the estimation of the coefficients of the regression model.

With respect to the climate: as mentioned as a response to a previous question: we consider the weather types as homogeneous in the study area and use these weather types in our regression model.

P7 L2: please include "river" before "density".

OK. Changed in the revised manuscript. "With respect to river density, the results show less clarity"

**Section 4.2 Effect of single drivers.**

P7 L3-11. The authors mention here the use of Wetspa model. However, nothing about it was mentioned before in the methodology section. Further, this reviewer does not understand very well the inclusion of that model. It seems the authors try to justify or explain the results obtained in their research with the results of a model that they do not explain how has been calibrated and applied in the study area. The model was applied by the authors in the study area? In any case, is it really necessary to include this information in the paper? If so, the authors should explain a bit more on its application and justify the need of it. In addition, the manuscript already has lots of figures to add new ones. This part of the section is confusing for this reviewer. "Using runoff coefficient as a proxy for peak flow anomalies", the authors should include some reference for this.

We did not calibrate, nor apply, the Wetspa model for our 29 catchments. We merely use the concept of potential runoff coefficient as used in the Wetspa model structure. This is now stated more clearly in the manuscript by adding the word 'structure' in the following sentence: "These findings correspond to an analysis done on the potential runoff coefficient as used in the hydrological model structure Wetspa (Liu and De Smedt, 2004)".

In this model structure, a potential runoff coefficient is defined as the ratio of runoff volume to rainfall volume; for estimation of these runoff coefficients, reference values are taken from literature (Browne, 1990; Chow et al., 1988 and Fetter, 1980). We compare our results, which are data-based, with the runoff coefficients from a hydrological model structure. As such, we do not feel the need to include Wetspa in the methodology section.

And, yes, according to us it is necessary to include this here, as this comparison confirms our findings.

"Using runoff coefficient as a proxy for peak flow anomalies" => to us, this goes without reference: a higher potential runoff coefficient is equal to higher runoff volume (considering equal rainfall amounts). And it is self-evident that higher runoff volumes will cause higher peak flows and consequently higher peak-flow anomalies.

"From a hydrological point of view and with the above definition of potential runoff coefficient in mind, relative changes in this potential runoff coefficient […]".

Firstly slope and texture are mentioned. However, there I no discussion on them. The authors only try to justify their findings on the use of the Wetspa model.

The link between these catchment characteristics and runoff coefficient/peak-flow anomalies has been investigated before, in qualitative studies and has been quantified by model-based approaches such as the development of the hydrological model structure Wetspa. In those studies (see the various references on the Wetspa development), extensive literature review was done on their conceptualization of the runoff coefficient and tabulated this as a function of land use, soil texture and slope. Our study confirms these previous studies in a data-based way and we choose not to repeat the hydrological interpretation of these results.

Second, the climate system. Little is also said about this. The existence of a negative correlation between some weather types, could be related with the fact that, when one increases the other inevitably decreases? I might be wrong, however, this paragraph on climate system does not give many information and results confusing.

Yes and no. When one increases, only one of the remaining other weather types has to decrease – sometimes there is a positive correlation: NW and N have a positive correlation of 0.36. Added this second example of correlation in the manuscript: "for example frequencies of anticyclonic and cyclonic weather show a negative correlation of -0.79, and frequencies of NW and N have a positive correlation of 0.36."

Finally, when considering land use, the authors only consider urbanization. If this is their main goal in the paper, it should be clear from the title and the introduction.

Urbanization is not the main goal of this paper, but only a part of it. We still investigate the other LULC changes, mainly when looking at the interaction effects.

**4.3 Interaction effects.**

This reviewer really appreciates the effort made including this new sub-section.

In figure 12 c) only one slope rage is considered, so it is difficult making the same comparison as in the other figures.

Note this is now subfigure (d) instead (c). This needs to be seen as follows: in catchments with a low loamic content, the effect of the slope on peak flow anomalies is lower, compared to catchments high in loamic content.

Here, I have the same questions on Wetspa model that I had in section 4.2.

See answer on this comment before.

**Others:**

P2 L26 and P3 L21. As suggested by one of the other reviewers before, brackets should be deleted and the authors should decide if they want to maintain or delete the words "relatively" and "main".

OK. Deleted the brackets of relatively and main in the revised manuscript. Probably the reviewer is referring to P3 L26 and P2 L21 of the track changes version of the latest revision – note that the brackets on P3, L21 were already removed in the previous revision.

P3 L21-24: As suggested for the introduction section by one of the other reviewers before, "Sect." should be replaced by section. Please revise all the text in this sense.

OK. Replaced Sect. by Section throughout the manuscript

There is no need for writing "+" signal before increasing percentages, specially, when it is clearly said in the text that the trends are increasing. So please delete those signals all over the text and abstract: For example. P1 L22: "precipitation might increase with +50 % in winter…" should be "precipitation might increase a 50 % in winter". Please check also for English correctness.

OK. Deleted the "+" signs, and replaced "might increase with" by "might increase by" for English correctness.

13 figures can be too many for a manuscript. The authors should try to summarize the information or select the really important ones and publish the rest as supplementary material.

Also see our reply on similar comment by the editor. We combined two figures, and moved some figures and tables to supplementary material, as suggested.

Decision: accepted to minor revisions.

It appears that the author undertook a substantial revision. The authors emphasize that both individual drivers and interaction terms are important in explaining the observed changes in river peak flows.

The main problem I had with the first submission was a lack of details and general conclusions. The problem was largely addressed in the revision, but I am not quite satisfied yet.

First, the abstract ends with a sentence saying interaction terms explain up to 32%. It appears to end again without a general conclusion. What is the implication of the interaction terms explaining 32%? Did the authors want to say interaction terms must be considered in such studies or something else? Then please say explicitly. After reading the abstract, my eyebrows went up and I thought "is this the end?"

We now end the abstract stating "This shows the importance to include such interaction terms in data-based attribution studies."

Second, there are still numerous areas that need minor changes. Hereby I list them from the beginning:

The authors use 'e.g.' too often. Most of them are quite annoying and 'for example' should be used instead.

Changed most of the 'e.g.' to 'for example'.

P1: Words 'our' appear a couple of times. I recommend removing/replacing them because the meaning is vague.

Changed "our study" to "this study", and "our findings" to "the findings of this study" in the manuscript.

Changed "our rivers" to "rivers worldwide" in the abstract.

P1L33: I am not sure why the colon is used

Rephrased to: "With respect to the attribution issue in the second step, it is noted that different drivers act in parallel in a complex hydrological system, with interactions between those drivers."

P2L7: "most studies hypothesized that deforestation…" I wonder whether the studies 'hypothesized' or actually 'found' that deforestation cause increased surface runoff. If they merely hypothesized, then what happened to the hypotheses?

Changed the "hypothesized" to "conclude". And, expanded on the literature review.

P2L26: "Section" and "Sect." are used mixed in the document

See also comment of reviewer 1. Changed throughout the manuscript.

Section 2: the authors added a section about data availability at the end. Therefore, all the URLs in this section seem unnecessary (and are annoying)

OK. Deleted the URLs in section 2.

P3L1: "Koppen" has an Umlaut

OK. Changed in the revised manuscript.

P3L4: "1 – 1.5" means one minus one point five. Should be "1-1.5"

Changed to "has increased over the past 30 years by 1 to 1.5 °C".

P3L24: "proposed by (Willems, 2009)" Please correct the citation format. There are other such cases in the document

OK. Corrected the citation format on this location, and other locations.

P4L15: the authors mix commas and semicolons in parentheses

Indeed, this is a typo. Has been corrected in the revised manuscript: "W; (NW, N); (NE, E, SE); (S, SW); U; C; A"

P5L20: "on (100 times) 20 random" I don't understand what the authors meant to say

We tested 100 linear models, each time based on 20 different random calibration catchments. We deleted "(100 times)" in the revised manuscript for clarity.

P6L2: "17.85*Sediment" I wonder why sediment is here. In addition, in this sentence, what did you mean by "the coefficient becomes –3.04 – 0.85*Slope…"? Did you put 1 for Sediment? Then again why Sediment is left in the last term?

Note these examples talk about Settlement and not Sediment.

Sediment should not have been in the last term, this is a typo and should have been A for Anticyclonic. This is now corrected in the revised manuscript.

We do not put 1 for Settlement; we are talking about the overall coefficient for Settlement in the final regression model and thus the exact percentage of Settlement is of no importance here.

P6L5: I don't see how Figure 7 shows that the model explains 60% of the changes. Seeing the figure again, I still don't get it.

Changed the reference to the figures: "The model as shown in **Error! Reference source not found.** is able to explain 60% of the changes in river peak flows over time (**Error! Reference source not found.**)."

P7L2: I don't understand "we do make any statements…"

This should be: "we do not make any statements …" Corrected this in the revised manuscript.

P7L12-13: 10%+6%+6% = 22%. Does it mean that there is additional 10% you did not further explain?

Yes, correct. That is why we state "[…] is largely carried by three terms only". (1st paragraph of section 4.3)

LULC and climatic conditions (this is a 4th term) explain another 2% (3rd paragraph of section 4.3).

The remaining 8% is explained by other factors from Table 3, but is not further discussed (4th paragraph of section 4.3).

P7L18-19: authors say "with increasing slope,…" but Figure 12(a) shows decreasing slope from left to right. I recommend plotting the graph with increasing slope from left to right, consistent with (b) and (c).

Increasing slope goes from top to bottom, which is consistent with the loamic content in subplot (b) and in the other interaction plots. Hence, we prefer to leave this as is currently stands.

P8L28-29: "The model also showed that, for most of the considered case studies,…" What did you mean by the considered case studies? Are they previous studies? If so, how are they relevant here?

The considered case studies = the considered catchments in this study. Changed this in the revised manuscript: "The model also showed that, for most of the considered catchments in this study, a decrease in forested area to increase settlement area indeed leads to increased peak flows."

P8L14-16: I don't think these sentences are appropriate for the conclusion

OK. Moved this paragraph to the final paragraph of section 4.1.

[revised manuscript text omitted]